# SEMANTIC-METRIC BAYESIAN RISK FIELDS: LEARNING ROBOT SAFETY FROM HUMAN VIDEOS WITH A VLM PRIOR

## ABSTRACT

Humans interpret safety not as a binary signal but as a continuous, context- and spatially-dependent notion of risk. While risk is subjective, humans form rational mental models that guide action selection in dynamic environments. This work proposes a **framework** for extracting implicit human risk models by introducing a novel, semantically-conditioned and spatially-varying parametrization of risk, supervised directly from **safe** human demonstration videos and VLM common sense. Notably, we define risk through a **Bayesian** formulation. The prior is furnished by a pretrained vision-language model. In order to encourage the risk estimate to be more human aligned, a likelihood function modulates the prior to produce a relative metric of risk. Specifically, the likelihood is a learned ViT that maps pretrained features (e.g., DINOv3), to pixel-aligned risk values. Our pipeline ingests RGB images (i.e. scene objects) and a query object string (i.e. the manipulated object), producing pixel-dense risk images. These images that can then be used as value-predictors in robot planning tasks or be projected into 3D for use in conventional trajectory optimization to produce human-like motion. This learned mapping enables generalization to novel objects and contexts, and has the potential to scale to much larger training datasets. In particular, the Bayesian framework that is introduced enables fast adaptation of our model to additional observations or common sense rules. We demonstrate that our proposed framework produces contextual risk that aligns with human preferences. Additionally, we illustrate several downstream applications of the model; as a value learner for visuomotor planners or in conjunction with a classical trajectory optimization algorithm. Our results suggest that the proposed method is a significant step toward enabling autonomous systems to internalize human-like risk reasoning. Additional visualizations and a link to our codebase can be found on our website.

## 1 INTRODUCTION

In safety-critical systems, traditional safety specifications are often grounded in precise physical models and formal mathematical constraints. However, many real-world scenarios involve safety considerations that are inherently semantic, context-dependent, and difficult to formalize—what is referred to as *intangible safety* or *contextual risk*. These specifications are not easily captured by equations or physical laws but are instead defined through semantics, human intuition, or data-driven insights. The goal of this work is to formalize *risk* in a statistically rigorous framework, demonstrate how *risk* can be regressed from common sources of data without explicit additional human labeling, and illustrate possible use cases for our proposed *risk* model.

*Risk* encompasses a range of scenarios where safety cannot be strictly defined by physical parameters alone. For instance, ensuring that a robot does not spill the contents of a container requires understanding the context of the task and the properties of the object, which may not be fully captured by physical models. Similarly, manipulating sharp objects around humans necessitates a nuanced understanding of human-robot interaction, beyond mere collision avoidance. These examples highlight the need for safety specifications that account for semantic understanding and contextual awareness.

The specification of *risk* is often data-driven, relying on demonstrations, sensor data, and learned behaviors. Techniques such as learning from demonstration, semantic mapping, and natural language processing enable systems to infer safety constraints from human input and environmental context. For example, a robot may learn safe handling procedures for sharp tools by observing human demonstrations, capturing the subtleties of human caution and care. Similarly, semantic understanding of objects allows models to generalize across diverse images and object interactions.

Despite the growing importance of *risk*, there is a lack of formal frameworks to model and guarantee such safety specifications. Traditional control-theoretic approaches may fall short in capturing the semantic nuances and contextual dependencies inherent in *risk*. Therefore, there is a pressing need to develop new methodologies that integrate semantic understanding, data-driven learning from readily available sources, and uncertainty quantification to ensure safety in complex, real-world scenarios, especially in human-robot interactions.

In this work, we introduce a novel framework for modeling risk and encourage risk-aware behavior by generalizing traditional set-based safety constraints into a continuous, object-centric risk representation. While prior approaches often define safety in terms of binary constraints—categorizing states as either safe or unsafe—our method captures safety as a scalar risk field varying across space, assigning a continuous value to each state. Like humans, the absolute value of *risk* has no physical meaning, but there exists an implicit *metric* in which to compare different object interactions. Colloquially, if **A** is riskier than **B**, and **B** is riskier than **C**, then **A** is riskier than **C**. Moreover, for common objects and scenarios, the risk between two objects is a non-decreasing function of their distance. Under these broad rules, the regressed scalar field enables a more nuanced understanding of safety, allowing robots to assess varying degrees of risk associated with different actions and environmental contexts.

Our approach draws inspiration from human behavior, where safety assessments are often context-dependent and object-centric. For example, humans instinctively exercise greater caution when handling a glass of water near electronic equipment compared to when they handle an empty cereal box. To emulate this behavior, our framework evaluates safety with respect to specific objects in the environment, enabling robots to generate trajectories that maintain appropriate distances from objects based on their associated risk levels. Moreover, semantic feature models have been increasingly powerful (Siméoni et al., 2025), enabling models operating on this feature space to display human-like understanding of images and language.

We develop an easy-to-interpret and expressive parametric risk model that integrates sensory data contextual information, and common sense to estimate the risk associated with various objects and scenarios. This risk model can then be used for downstream planning tasks. For instance, it can implicitly serve as a value function for trajectory optimization using world models or explicitly as a scalar field for traditional trajectory optimizers.

We validate our approach through a series of tabletop manipulation experiments involving a mixture of objects with varying risk profiles. Compared to state-of-the-art learning-based policies trained on risk-aware data, trajectories derived from our risk model are more risk-aware and make context-sensitive decisions, leading to safer and more efficient operation despite using a more naive planning method. Furthermore, we demonstrate that our risk framework, although partially derived from VLMs, outperforms state-of-the-art VLMs in producing human-aligned risk assessments of trajectories. Finally, we validate that the posterior model is greater than its parts, exhibiting better human alignment than both the likelihood and the prior.

## 2 RELATED WORK

### 2.1 SEMANTIC SAFETY

Traditional robot safety approaches often focus on physical constraints (e.g. collision avoidance or joint limits) but overlook higher-level semantic context. Semantic safety refers to safety considerations based on an understanding of object and task semantics or human intentions, rather than just distances and forces. Recently, researchers have started integrating semantic understanding into safety frameworks. For example, Brunke et al. (2025) introduce a semantic safety filter that leverages the contextual reasoning capabilities of LLMs to infer semantically unsafe conditions in a 3D semantic map, thereby imposing "common sense" and practical semantic constraints that go beyond geometry.

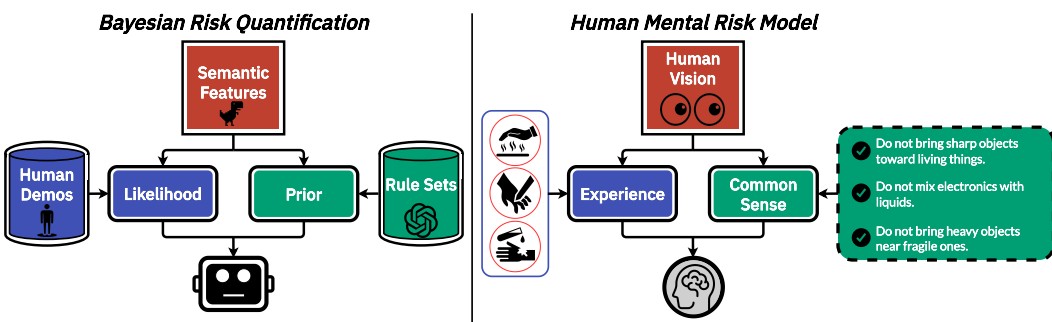

Figure 1: Bayesian risk quantification attempts to model implicit human mental risk reasoning specifically in regards to pair-wise object interactions *without* unsafe examples such as manipulating knives near people or water above electronics. Risk depends on distance and object pair semantics. These two effects can be learned respectively through two different data streams: fine-grained life-experience or resource-intensive human demonstrations and coarse data derived from simple human queries or knowledge from foundational models. This decomposition of risk aligns well with a Bayesian interpretation, where risk corresponds to a posterior derived from a human data-driven, distance-based likelihood and a machine-driven, semantics-based common-sense prior.

In a similar vein, Nakamura et al. (2025) propose a latent safety filter learned from unsafe demonstrations to avoid entering *intangible unsafe* regions at test time. Other work has explored incorporating semantic context and human feedback directly into safety models. For example, Santos et al. (2025) leverage Vision Language Models (VLMs) to interpret language instructions to continually update a robot's safety constraints during deployment. These works motivate the importance of semantic context and human knowledge in defining safety beyond simple physical constraints.

## 2.2 LARGE LANGUAGE MODELS IN ROBOTICS

Large Language Models (LLMs) have seen a surge in popularity for use in robotics as a way of translating human instructions into high-level actions for the robot to perform. SayCan (Brohan et al., 2022) proposes an end-to-end pipeline that translates problem-based text into a sequence of robot actions to perform to resolve the problem. Subsequent works have scaled up this idea: Driess et al. (2023) presented PaLM-E, a multimodal model that incorporates visual and state inputs into an LLM, which returns robot actions for open-world navigation and manipulation. For example, Text2Motion (Lin et al., 2023) combines an LLM planner with a geometric feasibility checker. Recently, vision-language-action models like RT-2 (Brohan et al., 2023) demonstrate that web-scale vision–language knowledge enables robots to recognize novel objects and actions by name. Other researchers have explored the use of LLMs for high-level decision making, paired with low-level controllers. For instance, Huang et al. (2022) demonstrated that frozen LLMs can be used as zero-shot planners by prompting them with available actions and environmental context. Likewise, ELLMER (Mon-Williams & colleagues, 2025) is a GPT-4–based framework that generates structured Python code for robot tasks, while execution relies on force and visual feedback, keeping the LLM in the loop only for high-level updates. Firoozi et al. (2025) provides a comprehensive overview of foundation models in robotics, demonstrating impressive common-sense reasoning capabilities of these models while highlighting reliability issues stemming from unsafe hallucinations. To counteract these issues, Ren et al. (2023) propose calibrating uncertainty and having robots proactively ask for human help when confidence is low. We believe that our work can boost the performance of foundation-level policies by providing semantically-conditioned, human-aligned risk signals both at train and test time.

## 2.3 RISK ESTIMATION FROM DEMONSTRATIONS

Learning cost or risk functions from demonstration data is a long-standing goal in imitation learning and inverse reinforcement learning (IRL). Classic IRL works (Ng & Russell, 2000; Abbeel & Ng, 2004) showed the recovery of reward functions that are optimally aligned with demonstrations. Recently, others have worked on inferring safety-related costs or constraints from demonstrations. Brown et al. (2020) introduced Bayesian Reward Extrapolation (Bayesian REX), which learns a

reward function from preference-labeled demonstrations and importantly quantifies uncertainty over that learned reward. By maintaining a posterior distribution over possible rewards, their method enables reasoning about risk (e.g., probability that a policy is unsafe) and provides confidence intervals on policy performance. Beyond rewards, others have aimed to infer constraints or risk measures from demonstrations. Chaubey et al. (2024) learn both a cost function and explicit constraints by carefully analyzing only safe trajectories. Our method relies on a Bayesian framework to model uncertainty and incorporate priors, inferring risk from regions avoided by humans and from subtle execution cues (e.g., detours). A key difference between the proposed method and prior work is our use of semantic context via pixel-aligned VLM features and an LLM-derived prior over pairwise object interactions. By regressing a dense spatial risk field at the sensor resolution, we provide fine-grained risk estimates across the scene, producing rich inputs for downstream planning.

## 3 RISK AND VIABILITY AS A BAYESIAN RANDOM VARIABLE

Given a context $\phi$ (e.g. semantics of interacting objects) and the distance $d$ between these two objects, we define the *viability* $v$ as the conditional distribution of the event of *being safe* conditioned on the context and the binary event of the distance being below a certain threshold

$$v(d, \phi) = \alpha(d)P(safe|\hat{d} < d, \phi) \geq 0, \tag{1}$$

where $\alpha$ is a scaling factor and non-decreasing in distance. We show later that we can find a suitable $\alpha$ that fits the definition (1). This degree of freedom accounts for viability belonging to a *family* of metrics to reflect subjectiveness (Theorem 1). Similarly, the risk follows suit, as demonstrated in Corollary 1.

**Theorem 1 (Viability Consistency)** *If $\alpha(d)$ and $P(safe|\hat{d} < d, \phi)$ are non-decreasing functions of distance, then $v(d, \phi)$ and $P(safe|\hat{d} < d, \phi)$ will display consistent rankings across $\phi$ for a constant $d$, and likewise across $d$ for constant $\phi$.*

**Corollary 1 (Risk Consistency)** *If the risk is a non-increasing function $f$ of the viability (i.e. $r = f \circ v$), then the risk displays anti-consistent rankings across $\phi$ for constant $d$, and similarly across $d$ for constant $\phi$. Therefore, operations like 1 minus the viability preserves trends between the risk and viability.*

While the safe conditional distribution is not a quantity that can be easily understood, it can be factored into simpler parts through Bayes Rule

$$P(safe|\hat{d} < d, \phi) = \frac{P(\hat{d} < d|safe, \phi)P(safe|\phi)}{P(\hat{d} < d|\phi)}. \tag{2}$$

The likelihood $g(d, \phi) = P(\hat{d} < d|safe, \phi)$ is the Cumulative Distribution Function of $p(d|safe, \phi)$, the distribution of distances given the context and the fact that the distance is safe. Colloqially, the likelihood is the probability that there exists a distance below some $d$ which is safe. This function is typically understood to be regressed from *observations* and applied to new observations. Specifically in this work, the likelihood will be regressed from human-derived videos involving *safe* object interactions.

The prior $h(\phi) = P(safe|\phi)$ is the probability of the two objects being safe regardless of their distance. In this work, the prior will be modeled through the immense common sense core contained within large language models (e.g. ChatGPT 5), allowing for vast generalization to different object interactions.

**Remark 1** *The normalization factor $P(\hat{d} < d|\phi)$ is **impossible** to compute, as it would require having access to **unsafe** data points. However, we show that it is unnecessary to compute the viability. Note that if the distribution of distances is* independent *of the context, the normalization factor satisfies the conditions for $\alpha(d)$. As a consequence, the viability can be defined simply as the product of the likelihood and prior*

$$v(d, \phi) = P(\hat{d} < d|safe, \phi)P(safe|\phi) \in [0, 1]. \tag{3}$$

Why should the distances be **independent** of the semantics? In manipulation, a robot can move a manipulable object anywhere in the workspace. Thus, without conditioning on the policy or the task, we should not assume that the distribution of distances should be biased for one set of semantics or another.

It is true that objects in reality do occur with other objects, or away from other objects, thereby inducing a distance distribution. However, we argue that this observation does not preclude the marginal distance distribution from being independent of semantics. In fact, the real life distance distributions are **conditional**, induced by a task or the implicit need to be safe. For example, plates are typically placed near to knives and stoves because they are used in kitchen-related tasks. In the same vein, books are typically not stored in the kitchen due to water damage. Note that the distance distribution conditioned on safety is the **likelihood**. Since the marginal distribution is not conditioned on task or safety, we find that the independence assumption is palatable for tractability reasons.

**Remark 2** *We find that the simple product can lead to overly conservative behavior. For example, if the prior predicts a value of 0 (unsafe), then the posterior will also be 0, regardless of the likelihood. To discourage conservatism, we modulate the prior based on the distance between the two objects. Specifically,*

$$P(safe|\phi) \leftarrow 1 - [\exp(-\lambda d)(1 - P(safe|\phi))], \qquad (4)$$

*with hyperparameter $\lambda$ responsible for tuning the influence of distance on the attenuation. In our work, we set $\lambda = 0.5$ and distance is measured in meters.*

We again emphasize that all models, especially the likelihood, are regressed from **safe** demonstrations. Therefore, the risk framework requires no unsafe or possibly unsafe examples, like bringing a knife close to a person or spilling water on books. This is in stark contrast to data-driven safe planning methods (Nakamura et al. (2025)), which prevents them from seeing deployment in truly risky scenarios.

## 4 LIKELIHOOD REGRESSION

The likelihood function maps pair-wise object contexts to parameters of a Cumulative Distribution Function. The objective of the likelihood is to properly capture the preferences of humans toward risk through real-life demonstrations. Because humans must actuate their preferences in the real-world for select combinations of context, the size of the dataset collected to regress this function is typically small and does not sufficiently cover all possible contexts that can occur in reality.

**Remark 3** *The **likelihood** is trained on a very small dataset and is itself a small transformer of 52 MB. In this work, we emphasize that the likelihood is only used to finetune and align the risk estimates for relevant objects of a particular test-time task. However, the use of DINOv3 features widens the net of objects (e.g. different colors, geometries, adjacent categories like "scissors" versus "shears") whose risk estimates can be accurately estimated. In spite of this fact, the **prior** is the main driver for generalization. As a result, the data collection process should be quick and simple, and the training of the likelihood should be fast. While the risk framework is amenable to larger datasets and larger networks, we leave the creation of a risk foundation model to future work.*

### 4.1 DATA COLLECTION

The likelihood function is regressed from a human participant performing simple pick and place tasks on a tabletop across various combinations of manipulated objects and central objects that are to be avoided. The participant was asked to maneuver the object safely around the central object. Seven RGB-D videos between 30 seconds to 2 minutes were collected using an Intel Realsense D435. More details about the human manipulation demonstrations can be found in Appendix A.

Our data processing pipeline converts these human demonstration videos into parameters of CDFs conditioned on object-pair semantics. First, we retrieve bounding boxes using YOLOv8 (Jocher et al., 2023), which are fed to SAM2 (Ravi et al., 2024) for object masks through time. The manipulated object mask is retrieved through HoistFormer (Narasimhaswamy et al., 2024).

The central object is defined as objects intersecting the straight line trajectory between the pick and place locations of the manipulated object. The SAM2 masks then allow the computation of

the manipulated-central object distances, which are tallied up throughout the entire video. These tallies form a semantics-conditioned histogram per trajectory (from pick to place). Each histogram has associated $L$ DINO features for both the manipulated and central object, forming $L^2$ pairwise combinations. Only 100 feature pairs are sampled per histogram and added to the training dataset.

Parts of the data processing pipeline are visualized in Figure 2. The cup (red) is the manipulated objected, while the stove that intersects the straight line path (yellow) is the central object. For pick and place tasks, we do not want the table influencing the risk estimates. As a result, we set the table to have no risk, indicated in yellow. From a video, a discrete semantics-conditioned histogram is generated. For computation reasons, a smooth Bézier curve is fit on top of the CDF. The parameters of the Bézier curve are used as targets for the likelihood model. This choice of output not only makes learning smoother, but also greatly lowers the dimensionality of the output, enabling better generalization.

We choose to use DINOv3 (Siméoni et al., 2025) features as representations of the context due to the model's strong semantic understanding of in-the-wild images. Each RGB frame is passed through the DINOv3 model to retrieve a pixel-dense feature image. Features are averaged over object masks per-frame, yielding more stable training results.

### 4.2 MODEL

The likelihood is a function that maps two DINO features (manipulated and pixel) to control points ($N = 10$) of a Bézier curve, constrained to be between 0 and 1. Specifically, the function is a multi-headed self-attention transformer with 8 heads. The model is trained with a permutation-invariant structure to conform with the assumptions of an interchangeable risk field. For a single trajectory $\tau$, our training loss is a simple MSE loss.

## 5 PRIOR FITTING

The prior function maps pair-wise object contexts to a scalar probability of being safe. Unlike the likelihood, the prior is designed to *generally* understand human preferences across a wide breadth of contexts, which would otherwise be immensely time-consuming to *exactly* quantify through actual human demonstrations. To this end, we leverage the strong common sense knowledge embedded in models trained on internet-scale data, such as large language models (e.g. ChatGPT 5).

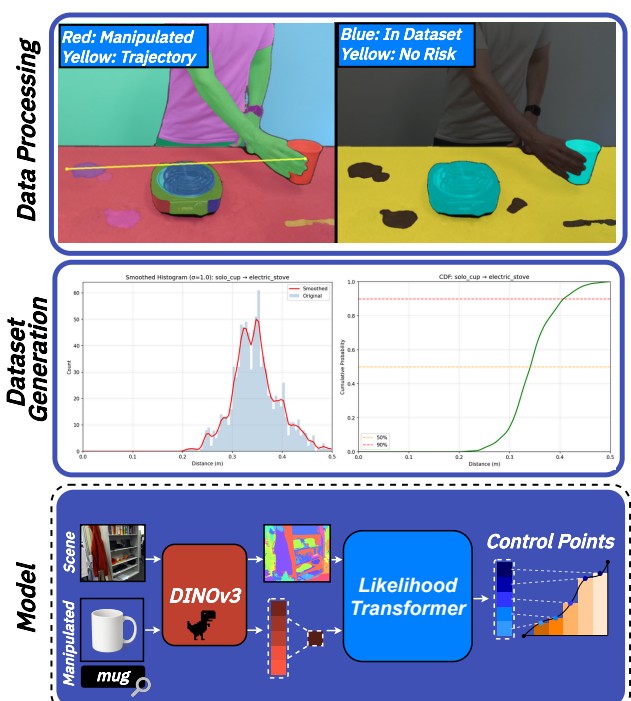

Figure 2: Top: RGB-D human demonstration videos are segmented and tracked. Middle: Per-trajectory inter-object distance histograms and their CDFs are extracted from the video streams. Bottom: DINOv3 features from the manipulated object and the scene are fed to the likelihood, predicts Bézier control points of a CDF.

Rather than learning human preferences from observations, humans can use LLMs to sample-efficiently enforce rulesets on large classes of inter-object interactions (e.g. "sharp objects should stay away from sensitive objects" or "do not mix objects containing liquids with electronics"). In practice, our results do not require the LLM to be explicitly prompted with these rulesets. A prompt that is more reflective of the one used in our experiments can be found in Figure 3.

**Remark 4** *We emphasize that the prior allows the risk estimate to generalize to many different object categories. The use of DINOv3 features expands the number of objects whose risk can be accurately estimated. Human operators can also inject semantic rules and additional object categories into*

*the prior on-the-fly through the LLM. However, the prior is not necessarily human-aligned. The **likelihood** is the main driver for human-alignment for test-time objects.*

## 5.1 DATA COLLECTION

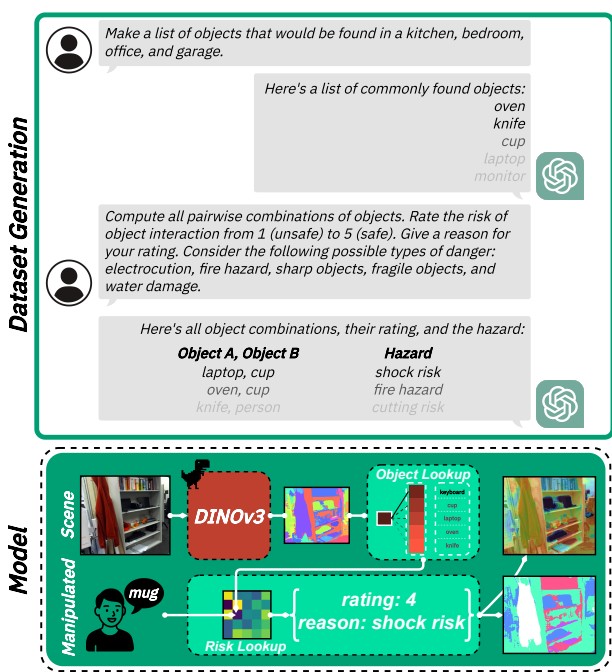

Figure 3: Dataset Generation: A prompt chain asks an LLM to list objects in particular settings, then score every pairwise combination for risk (1–5) with a hazard type and rationale. Model: Pixel features query an object lookup table for the nearest-neighbor object category. This label and the manipulated-object string is used in the risk lookup table to retrieve the correct risk ratings and reasoning.

Data is divided based on two different purposes: fitting of an object lookup table (LUT) and a risk lookup table. For the object LUT, ChatGPT is queried for a list of possible objects that can occur in desired settings (e.g. kitchens, bedrooms, offices, and garages). A representative image is generated for each object category. Specifically, internet images of the objects are obtained through a combination of automated search engine querying (e.g. Google Custom Search) and queries to the image-generator of ChatGPT. This large corpus of images is passed through DINOv3 to get pixel-dense latent features for each object. Subsequently, for every dense DINO image, we perform K-means ($K = 5$) to extract $K$ representative latent features per object category. The object LUT data contains 338 object categories queried across kitchens, bedrooms, offices, and garages. The full list of objects can be found in Appendix B.

Data for the risk LUT first requires the generation of an exhaustive list of pair-wise combinations (approximately 60K) of objects by the LLM. The LLM is then tasked with rating each pair of objects between 1 (unsafe) and 5 (safe), as well as the corresponding reason for the rating. To help guide the LLM toward human preferences, we ask it to select reasons from a pool of risks (e.g. "spillage", "crushing", "fire hazard", "electrocution"). Unlike humans, LLMs can perform this exercise on a massive scale extremely quickly, which can additionally be iterated with the help of a human. For example, more task-aligned behavior can be achieved by adding risk types and rulesets to the LLM query. As an example, for tabletop manipulation experiments, we ask the model to identify any interactions with a table-like object and set its associated risk rating to 5. However, for our experiments, no other rulesets were added to the query.

## 5.2 MODEL

Due to the discrete nature of the dataset format and the strong generalization power of DINO, a simple lookup table is a strong candidate for the prior. At small sizes, a lookup table is performant, efficient, and interpretable. Moreover, we do not need the model to inherently generalize well across unseen objects since we rely on LLMs as a data-generating function to collect a more diverse set of objects if needed.

The prior model consists of two lookup tables: an object LUT and a risk LUT. The object LUT performs a nearest-neighbor query to a DINO feature in the list of keys, returning the associated object category as a string. Specifically, the keys consist of $K$ features per object category corresponding

to the $K$ cluster centroids. We find that the use of K-means, as opposed to averaging, discourages spurious matches to different objects that may share a particular part (e.g. a human leg to a table leg).

The risk LUT is simply a dictionary, using the matched object string from the object LUT and the specified manipulated object string (e.g. "cup") to find the corresponding risk rating and reason produced by the LLM. All pixels in a scene image can be passed into the prior model to produce a pixel-dense risk and reason image in (Figure 3). For visualizations of the reason image, please see our website.

# 6 RESULTS

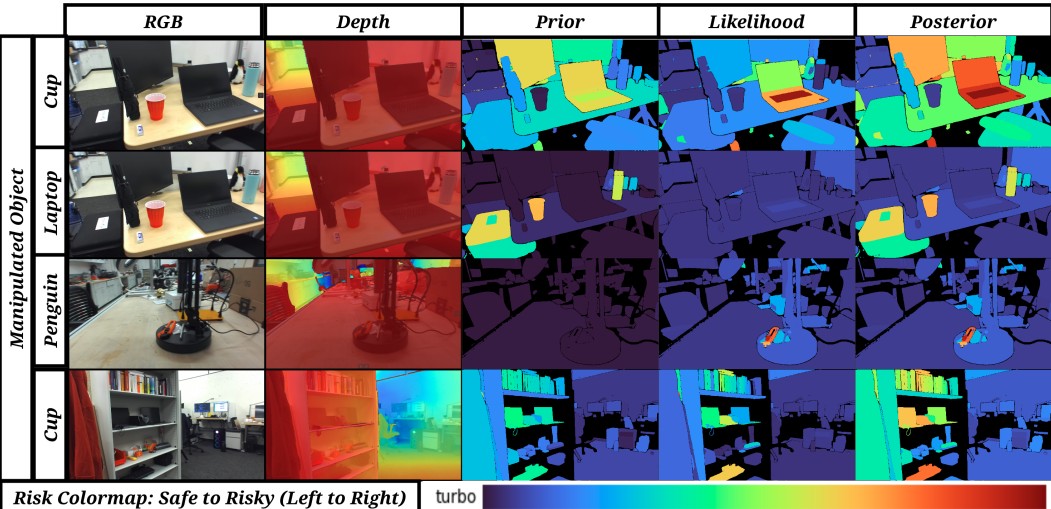

Figure 4: Ego-centric renders with corresponding manipulated object across different channels. Note that riskier objects are in red, while safer objects are in purple in accordance with the turbo colormap. In general, the posterior picks up risky objects from both the prior and likelihood, resulting in a more human-aligned risk image. More visualizations can be found on our website.

## 6.1 RISK QUALITY

The risk field is evaluated at each pixel (captured by a ZED Mini) with respect to a manipulated object in Figure 4. To produce cleaner images, we run SAM2 on the RGB channel and average the risk values over object masks. Overall, the posterior identifies risky objects while taking distance to objects into account. Many of the objects present were not in the training datasets, which speaks to the generalization of the risk framework. For example, the box cutter, keyboard, umbrella were not in the human demonstration. The penguin-cup example was in the human demonstrations. Subsequently, the cup-penguin pair lights up in the likelihood and posterior.

## 6.2 RISK AS A VALUE SIGNAL

Our risk framework produces a valid metric for comparing images and, in general, trajectories. This capability is shown in Figure 5, where the risk model is tasked with choosing the best shelf to place a "cup". Several video streams (one to each shelf) are passed through the risk framework, producing a stream of risk images. The riskiest shelf (4) contains a "laptop", followed by Shelf 2 containing a "keyboard". Shelf 3 is the least risky as predicted by our risk model and the best candidate to place the "cup" since there are no electronics or water damage risks. Additional visualizations are available on our website.

To quantify the human alignment of the risk estimate, we compare the risk rankings of the proposed risk framework and SOTA VLMs for several image streams in two environments: `Shelf` and `Kitchen`. The manipulated object was "cup". Specifically, each scene consists of 5 trajectories,

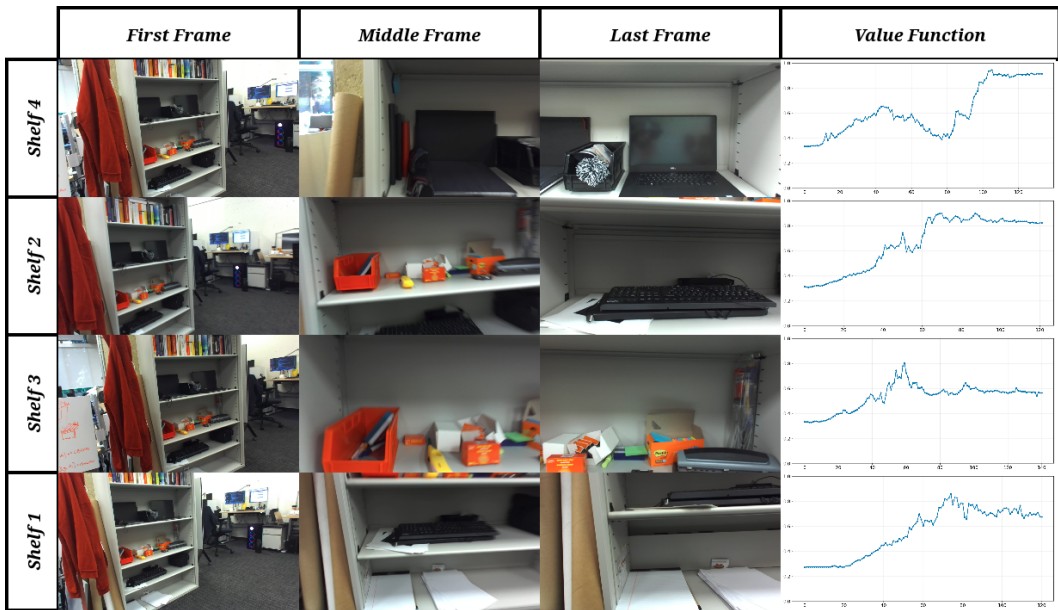

Figure 5: Comparison of four trajectories from the `Shelf` environment while manipulating "cup". For each trajectory, we show the first, middle, and last frames alongside the value function. The value function corresponds to the P75 (seventy-fifth percentile) of the per-frame risk distribution, providing a robust summary of trajectory risk. Higher risk shelves produce consistently elevated values, illustrating human alignment of the proposed risk framework.

with each represented as a video stream. Because the VLMs can only accept 10 images per prompt, we were limited to only the first and last frame of each trajectory. The VLMs were then asked to choose the least and most risky trajectory. Since our risk model is not limited by number of images (and thus can utilize more context), we pass in the first 10 and last 10 frames of each trajectory and retrieve the P75 risk curve (similar to Figure 5). The average risk across the frames was used as the risk of the entire trajectory. The trajectories were then sorted based on their average risk. 17 human participants were shown the full video stream of each trajectory and asked to rank the different trajectories. Additional details of the experiment can be found in Appendix C.

We evaluate our risk framework against SOTA VLMs (ChatGPT 5 and Gemini 3) on risk quantification, to show our method is more human aligned in Table 1. If the method's choice for least risky trajectory agreed with the human participants, then the method received +1. The same is true when rating the most risky trajectory. Because the VLMs are stochastic, they were queried multiple times to generate a representative sample. We find that our risk framework is human aligned while the VLMs can be often incorrect or exhibit high variance. The prompt for the VLMs can be found in Appendix D.

Table 1: **Human Alignment of Value Signals:** Human alignment of several methods on the least- and most-risky trajectory selections. + indicate "Thinking" versions of the VLM, while no marking indicates the "Instant" version. Because the VLM is stochastic, these models were queried multiple times. Methods who choose trajectories that agree with the human participants receive +1. Denominators indicate the number of queries. Our method achieves higher human alignment while being fully deterministic and faster than the more powerful "Thinking" VLMs. S = `Shelf` and K = `Kitchen`.

| Method | Least Risky (S\|K) ↑ | Most Risky (S\|K) ↑ |
|---|---|---|
| **Risk (Ours)** | **1/1 \| 1/1** | **1/1 \| 1/1** |
| GPT + | 2/3 \| 3/3 | 1/3 \| 3/3 |
| GPT | 4/5 \| 0/3 | 1/5 \| 0/3 |
| Gemini + | 0/3 \| 3/3 | 0/3 \| 3/3 |
| Gemini | 1/3 \| 1/3 | 0/3 \| 3/3 |

## 6.3 RISK-AWARE TRAJECTORY OPTIMIZATION

To quantitatively benchmark our risk framework, we compared the quality of trajectories produced by state-of-the-art robot policies trained on risk-aware demonstrations to a classical trajectory optimizer utilizing our spatial risk field. We trained Diffusion Policy (DP) (Chi et al., 2024) and fine-tuned GR00T (VLA) (NVIDIA et al., 2025) on approximately 30 human-teleoperated demonstrations, analogous to the data collected for training the likelihood model. An RGB-D image of the scene is passed through the risk model to produce control points representing the posterior as a function of distance. Given a certain viability threshold ($\alpha = 0.1$), a corresponding radius is extracted from the posterior function, which serves as a buffer radius around the depth point cloud. A classical trajectory optimizer (Chen et al., 2024) that navigates around a union of balls can be used to produce risk-aware and smooth trajectories. Additionally, a human teleops a risk-aware trajectory per experiment as a control.

For 16 robot experiments, the trajectories are rated by 14 human raters in terms of riskiness, which we report in terms of Top Box (T1B) and Top 2 Box (T2B) (2) out of 224 ratings per method. We find that the spatial risk field consistently produces more risk-aware trajectories than the SOTA robot policies. Moreover, we find that the risk model produces trajectories competitive with GR00T in similarity to the human control, despite using a more naive classical trajectory

Table 2: **Robot Trajectory Preferences:** Trajectory quality rankings and human-policy trajectory distances between a Vision-Language-Action model, diffusion policy, and our explicit risk-aware trajectory optimizer. Human demonstrations are provided as reference.

| Method | T1B ↑ | T2B ↑ | DTW (med./mean 95%) ↓ |
|---|---|---|---|
| VLA | 13 | 31 | 42.03 / **42.28** |
| DP | 14 | 29 | 47.81 / 48.57 |
| **Risk (Ours)** | **134** | **196** | **41.10** / 44.23 |
| Human | 62 | 191 | N/A |

optimizer. We measure trajectory similarity using dynamic time warping (DTW) and report the median and mean of the top $95\%$ of trajectories to filter outliers. Finally, in terms of T2B performance, the risk model returns trajectories that align with human preferences. More details can be found in Appendix E and videos of the robot trajectories can be found on our website. Certain objects like the drone were never seen in either the likelihood training dataset or the LLM output.

## 7 CONCLUSION

The proposed Bayesian risk framework is an interpretable and expressive risk model that transforms semantic information, human demonstrations, and common sense to predict a human-aligned risk metric. The proposed risk model can subsequently be used for relevant robotics downstream tasks like planning utilizing either learned and classical methods. In one instance, the image-based risk serves as a useful value function to choose risk-averse trajectories, such as those produced by stochastic policies or sampling methods used in conjunction with world models. On the other hand, since our risk model is also spatially aware, the risk field can be transformed directly into 3D and used jointly with classical trajectory optimizers. We demonstrate that the model produces risk-aware trajectories through a series of tabletop manipulation experiments featuring many combination of manipulated and scene objects. The results demonstrate that our contextual risk framework enables autonomous agents to make more risk-aware and context-sensitive decisions, leading to safer and more efficient operation.

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

## A    DATA COLLECTION PROCESS

There are two instances of data collection: for training the likelihood model and for training the baselines (i.e. diffusion policy and GR00T). Both are simple instances of tabletop manipulation of one object around another, or possibly on top if they are not risky. For the likelihood model, we only trained on videos of trajectories involving 7 distinct object interactions. Each video was typically 30 seconds to 2 minutes. Specifically, each video with 300 average frames were used to train the likelihood function, producing 648,000 training examples. For the baselines, a human tele-operated a manipulator holding an object and navigated it around another object, for approximately 30 different object interactions. The baseline data collection process took approximately an hour.

The human demonstrator is only instructed to grab an object and move it around another stationary object in a natural, non-risky way from a starting point to a goal point. For the likelihood data, the start and goal points are implicitly defined by where the person picks up and puts down the grasped object. For the tele-operated baseline data, the tele-operator similarly moves a pre-grasped object in a natural, non-risky way from a pre-defined start pose to a pre-defined goal location.

To create semantic-conditioned distance histograms, the inter-object distances are computed using the depth and object masks, tallied over a trajectory. A trajectory spans from when an object is picked up to when it is put down, as recorded by HoistFormer. Mathematically, for a trajectory of length $T$, our image processing pipeline returns $T$ pairs of latent vectors and inter-object distances $\{(m_t, o_t, d_t)\}_{t=1}^T$ for a particular context, due to the fact that the average Dino feature within an object mask changes slightly between frames. $\{d_t\}_{t=1}^T$ is used to create a 100-bin histogram for the trajectory. A discrete CDF is then extracted from the histogram (2).

## B    LIST OF OBJECTS AND OBJECT LOOKUP DETAILS

Below is the list of 338 object categories used in the lookup table. Accompanying each object category is an image automatically retrieved from the internet using Google Search Console. DINOv3 features are extracted from these images. Then, K-means ($K = 5$) is used to generate DINO feature centroids for each object category. We find that K-means is better for object detection than averaging features over the entire image. These centroids are inserted into the object lookup table, mapping to the same object category.

List of object categories:

action figure, adjustable wrench, air compressor, air fryer, air purifier, alarm clock, all-purpose cleaner, aluminum foil, angle finder, apron, av receiver, baking sheet, ball, bandage box, banister, bar clamp, barstool, bath brush, bath mat, bath towel, batteries, power bank, bed, lamp, belt sander, bench, binders, bit set, blanket, bleach bottle, blender, blinds, blocks, blow gun, board game, bottle, bolts, book, bookcase, bottle opener, bowl, box cutter, bread knife, broom, bubble wrap, bucket, bulletin board, bungee cords, business cards, c-clamp, cabinet, cable ties, calculator, calipers, can opener, candle, car battery, card reader, cardboard box, cards, caulk gun, chain, chair, changing table, chargers, chef knife, chisel, circular saw, cleaning wipes, clipboard, closet, coaster, coffee maker, coffee table, colander, comforter, compass, cooling rack, crayons, crib, crimpers, crowbar, cup, cupboard, curling iron, curtain, cutting board, degreaser, dehumidifier, deodorant stick, desk, desk fan, desk lamp, desktop pc, dice, dish rack, dish soap, dish towel, dishwasher, disinfectant, document trays, doll, dolly, door, doorknob, doormat, dremel, dresser, drill, dry erase eraser, dry erase markers, dryer, duct tape, dustpan, duvet, e-reader, electric kettle, electric shaver, envelope, erasers, espresso machine, extension cord, external hard drive, fan, faucet, filing cabinet, first-aid kit, flashlight, flat iron, floor mat, food processor, freezer, french press, frying pan, game console, game controller, garden hose, glass bottle, glass jar, glasses case, glue stick, grass shears, grater, hacksaw, hair dryer, hammer, hamper, hand saw, hangers, harness, headlamp, headphones, heat gun, hedge trimmer, hex key set, hoe, hole punch, hose, hot glue gun, humidifier, ice cube tray, impact driver, index cards, ironing board, jar opener, kettle, keyboard, keys, kitchen scale, knife, knife block, ladder, ladle, laptop, laundry basket, lawn mower, leaf blower, light switch, lighter, loofah, magazines, mallet, markers, matchbox, mattress, measuring cups, measuring spoons, mechanical pencil, media remote, medicine bottle, microphone, microwave, milk crate, milk jug, mirror, modem, monitor, mop, mousepad, mouthwash cup, muffin tin, mug, multi-tool, multimeter, nails, napkin holder, needle-nose pliers, nightstand, notebooks, notepads, office chair, orbital sander, ottoman, oven, oven mitt, paint

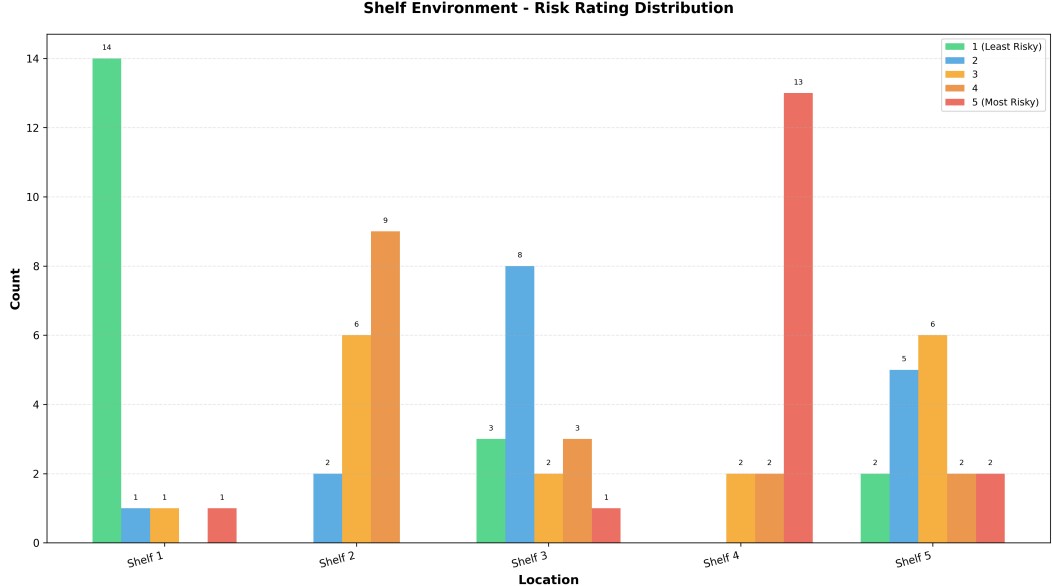

Figure 6: Distribution of ratings made by 17 human participants for different trajectories in the `Shelf` environment.

brushes, paint can, paint scraper, paint stir stick, paint tray, pan lid, paper bag, paper plate, paper towel roll, paring knife, peeler, pegboard, pencils, pens, phone charger, picture frame, pillow, pitcher, pitchfork, plant pot, planter, plastic bag, plastic bottle, plate, pliers set, plunger, pot, pot holder, potato masher, power strip, precision screwdriver set, printer, protractor, pruning shears, railing, rake, ratchet straps, razor, reciprocating saw, recycling bin, refrigerator, remote control, rice cooker, rivet gun, roasting pan, rolling pin, rope, router, rubber mallet, rug, ruler, salt shaker, saucepan, sawhorse, scissors, screwdriver set, screws, shop-vac, shovel, shower curtain, sieve, sink, smartphone, socket wrench, soda can, sofa, soldering iron, solo cup, soundbar, space heater, spatula, speakers, spice jar, sponge, spray bottle, squeaky toy, stapler, step stool, stool, storage bins, stove, stuffed animal, stuffed penguin, subwoofer, surge protector, table, tablet, tape measure, teapot, thermometer, thermos, toaster, toilet brush, toilet paper holder, toilet paper roll, tongs, toothbrush, toothpaste, toy car, trash can, tupperware, tv, tv stand, umbrella, usb flash drive, vacuum, vase, vice-grips, vise, wall clock, wardrobe, washing machine, water bottle, webcam, wheelbarrow, whisk, whiteboard, wine glass, wire cutters, wok, wooden spoon, work bench, wrench set

## C    VALUE LEARNER RATINGS

The anonymous questionnaire and the trajectory videos used in this experiment can be found on our website. The ratings from 17 human participants are tabulated in Figure 6 and Figure 7 for the `Shelf` and `Kitchen` environments, respectively. The VLM ratings are tabulated in Table 3. For reference, the `Shelf` trajectories are given the following IDs: Notepad (1), Keyboard (2), Stationery (3), Laptop (4), Books (5). The `Kitchen` trajectories are given the following IDs: Coffee Maker (1), Fridge (2), Laptop (3), Microwave (4), Monitor (5). For both environments, the manipulated object was "solo cup".

For the `Shelf` scene, our risk model rated the following trajectories in increasing risk order with risk value: Stationery (3: 0.4498), Notepad (1: 0.4818), Books (5: 0.5253), Keyboard (2: 0.5708), Laptop (4: 0.6245).

For the `Kitchen` scene, our risk model rated the following trajectories in increasing risk order: Fridge (2: 0.2618), Monitor (5: 0.4841), Microwave (4: 0.5180), Coffee Maker (1: 0.5888), Laptop (3: 0.5994).

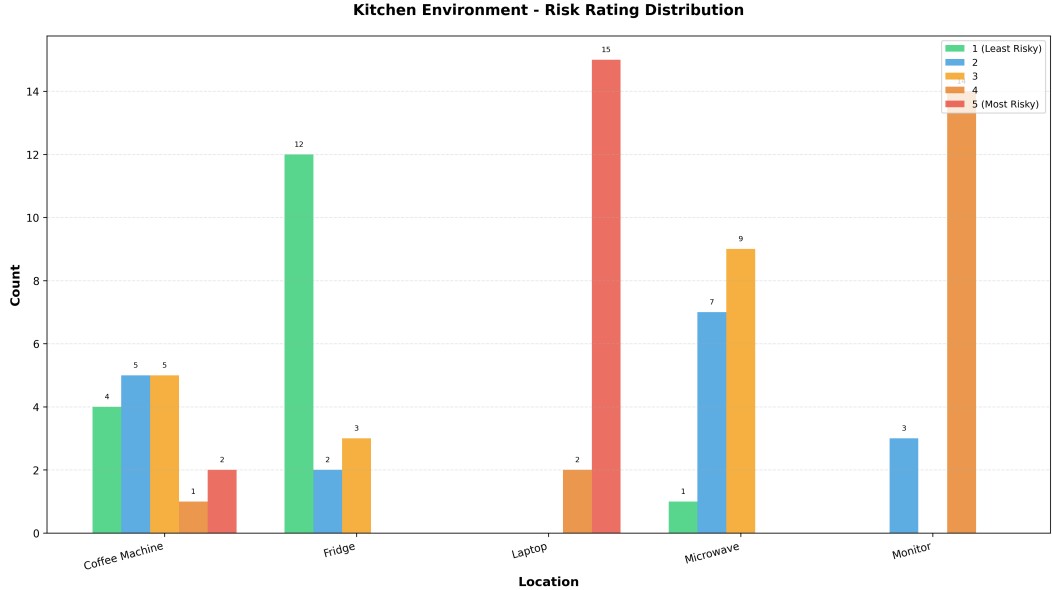

Figure 7: Distribution of ratings made by 17 human participants for different trajectories in the `Kitchen` environment.

| Method | Scene | Least Risky | Most Risky |
|---|---|---|---|
| GPT Thinking | Shelf | 3 | 1 |
| GPT Thinking | Shelf | 1 | 5 |
| GPT Thinking | Shelf | 1 | 4 |
| GPT Instant | Shelf | 1 | 4 |
| GPT Instant | Shelf | 3 | 1 |
| GPT Instant | Shelf | 1 | 5 |
| GPT Instant | Shelf | 1 | 3 |
| GPT Instant | Shelf | 1 | 5 |
| Gemini Thinking | Shelf | 4 | 5 |
| Gemini Thinking | Shelf | 3 | 2 |
| Gemini Thinking | Shelf | 5 | 1 |
| Gemini Instant | Shelf | 4 | 3 |
| Gemini Instant | Shelf | 1 | 3 |
| Gemini Instant | Shelf | 5 | 3 |
| GPT Thinking | Kitchen | 2 | 3 |
| GPT Thinking | Kitchen | 2 | 3 |
| GPT Thinking | Kitchen | 2 | 3 |
| GPT Instant | Kitchen | 1 | 5 |
| GPT Instant | Kitchen | 1 | 5 |
| GPT Instant | Kitchen | 1 | 5 |
| Gemini Thinking | Kitchen | 2 | 3 |
| Gemini Thinking | Kitchen | 2 | 3 |
| Gemini Thinking | Kitchen | 2 | 3 |
| Gemini Instant | Kitchen | 2 | 3 |
| Gemini Instant | Kitchen | 4 | 3 |
| Gemini Instant | Kitchen | 4 | 3 |

Table 3: Trajectory risk assessment by VLMs. Trajectories are represented by their IDs.

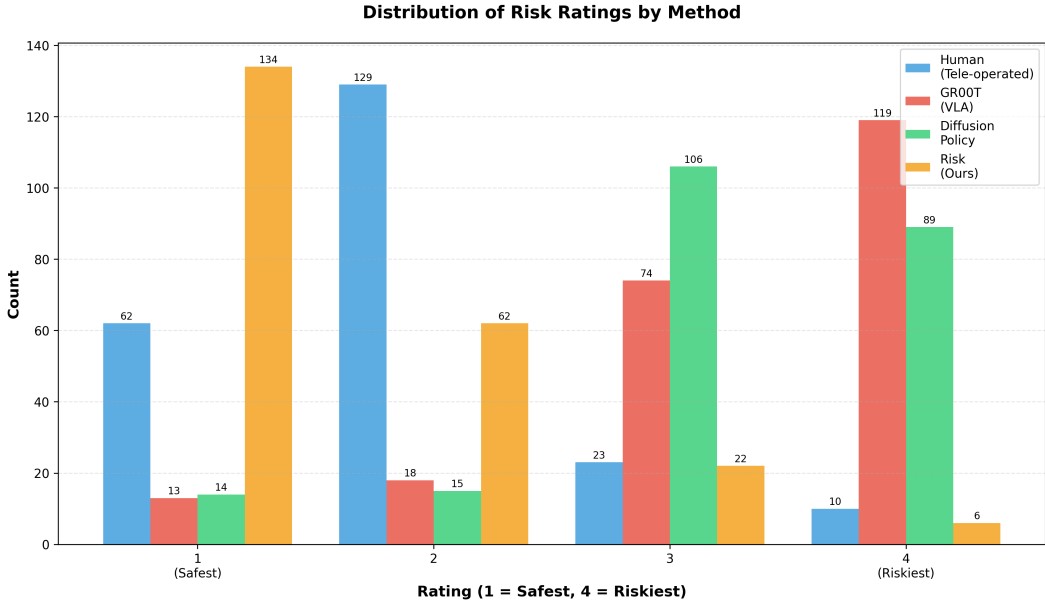

Figure 8: The distribution of robot trajectory ratings for 14 human raters and 16 robot scenarios.

## D    VALUE LEARNER PROMPT

We use the following prompt for the VLMs:

> Five trajectories are given, with the start and ending image frame of each trajectory inserted sequentially. The images are labeled as TrajID-FrameID.png. Assume you are a robot holding a cup of water. Which trajectory is the least risky, and which trajectory is the most risky?

## E    TRAJECTORY OPTIMIZATION RATINGS

The anonymous questionnaire and videos of the robot trajectories used in this experiment can be found on our website. The ratings are tabulated in Figure 8.

