# OpenReview forum: "Semantic-metric Bayesian Risk Fields: Learning Robot Safety From Human Videos With a VLM Prior"
_ICLR.cc/2026/Conference — Submitted to ICLR 2026_

### Official Review · Reviewer_uqYx · 2025-10-21

**Soundness:** 2
**Presentation:** 3
**Contribution:** 2
**Rating:** 4
**Confidence:** 3

**Summary:**

The paper proposes Semantic-Metric Bayesian Risk Fields, a framework that regresses pixel-dense, context-conditioned risk maps from safe-only human demonstrations and a VLM/LLM-derived semantic prior. The core modeling choice decomposes a “viability” measure into a Bayesian likelihood over distances conditioned on context (learned from RGB-D videos) and a prior over pairwise object semantics (queried and post-processed from an LLM), with the likelihood parameterized as a smooth CDF (via Bézier control points). The resulting risk maps can be used either as value signals for policies or for classical trajectory optimization. Experiments on tabletop manipulation suggest the approach produces qualitatively reasonable risk maps and, as a planner cost, yields trajectories that are rated safer than those from trained visuomotor baselines in the reported setup.

**Strengths:**

The formulation connects an intuitive decomposition to a practical pipeline that yields dense, usable risk fields with minimal bespoke supervision. Parameterizing the likelihood as a smooth CDF via low-dimensional Bézier control points is a neat design that regularizes the estimator and stabilizes training from limited demonstrations while enforcing monotonicity in distance by construction. The system-level integration is clear: DINOv3 features provide pixel-aligned semantics, the LLM-derived prior encodes coarse common sense at scale, and the posterior is consumed by both learning-based policies and a classical trajectory optimizer. Qualitative visualizations show that the principle works, are context sensitive to the manipulated object, and the trajectory optimization results demonstrate that the proposed explicit risk fields can be preferred over end-to-end policies in the reported tasks.

The figures are mostly clear and aid in understanding the main text.

**Weaknesses:**

## Contibution
The theoretical component is thin and rests on strong assumptions that limit real-world applicability. Methodologically, the approach largely assembles known components (SAM2/YOLO for masks, DINOv3 features, transformer on features, LLM-generated priors), and novelty resides mainly in the particular Bayesian factorization. Ablations quantifying the contribution of each choice are limited. The likelihood is trained from only seven safe-only videos, and the large count of “examples” stems from frame and pairing combinatorics rather than independent, diverse contexts. It is unclear how well the method would generalize to other scenes.

## Correctness
The key assumption that the distribution of distances does not depend on the context and the normalization factor can be left out, justified by “arbitrary” distance distributions or by experimental design, is not credible for most scenarios, where distances and motion are heavily context-, affordance-, and task-dependent. Researchers designing the distribution of distances for experiments will always choose distributions based on the task.

The central claim that risk is non-decreasing with distance is often violated in practice. Safety semantics go well beyond distances as orientation, velocities, occlusions, containment, support relations etc may all play a role. Theorem 1 amounts to a monotonicity/ranking statement under monotone mappings and adds little substantive guarantee. The introduction’s suggestion that the method “leverages guarantees from rigorous control-theoretic approaches” overstates what is proven. There is no guarantee for anything in the pipeline, especially since the inputs for calculating the semantic-metric bayesian risk fields come from black box neural networks.

The treatment of interchangeability is also questionable. Risk is framed as permutation-invariant in parts of the model, yet many pairwise risks are asymmetric with respect to which object is manipulated (e.g., moving water above a laptop versus moving a laptop near a stationary glass). This asymmetry should be reflected in how likelihoods and priors are conditioned.

The prior relies on LLM ratings that are themselves subjective and prompt-sensitive. The paper notes some manual rules (e.g., forcing tabletops to zero risk), suggesting non-trivial tuning for each scenario. Important implementation details are underspecified: how 3D distances are computed robustly from RGB-D streams (calibration, missing depth, occlusions), how the “straight-line” path is justified in clutter with obstacles unrelated to the focal pair’s semantics, runtime to generate risk fields, and how thresholds for viability-to-radius conversion are selected.

## Experiments
On evaluation, the baselines underperform strikingly (0/33 top-1 preferences for both VLA and diffusion policy), which raises concerns about setup, training parity, and metric sensitivity. Human ratings of “riskiness” are inherently subjective. The paper would benefit from objective proxies (e.g., collision/near-miss rates under perturbations, task success under constraints, or compliance with specified semantic rules). Claims of generalization to unseen objects/contexts are not rigorously substantiated beyond qualitative figures. Distribution shift tests, cross-scene generalization, and robustness to prior misspecification are missing.

## Language
Overall the paper is decently organized and the language is fair, but there are a several places where it requires polishing.
- On line 236 it's hard to understand what is meant by safe only human demonstrations.
- Other sections are missing words, contain overly long sentences, or are grammatically incorrect (e.g. line 238, 263, 384, 421)
- Active citations are being used as passive ones at line 431, 451, 456.

There are terminology and notation issues:
- “functional” is misused. A functional by definition refers to a mapping from functions as input to some scalar value. In the paper, the "likelihood functional" maps vectors to function parameters.
- The symbol d is overloaded for both distance and the number of frames/samples.

**Questions:**

- What concrete safety guarantees does the framework provide? Beyond the monotonic ranking lemma, can any bound or certification (e.g., chance constraints under model uncertainty) be stated for the induced trajectories?
- How sensitive are results to the distance-independence assumption? If $P(\hat{d} \leq d | \phi)$ depends on $\phi$, what changes in the estimator or the training objective, and does Theorem 1 still hold?
- How do you incorporate non-distance state variables (velocity, orientation, containment/support, spillage direction) that can dominate semantic risk? Can the likelihood be extended to a richer metric space while preserving smooth CDF parameterization?
- How is the LLM prior calibrated against human preferences beyond prompt engineering? Can you quantify inter-rater reliability, prior uncertainty, and posterior sensitivity to prior misspecification?
- Have you evaluated on unseen scenes, object categories, and camera viewpoints with held-out environments? The paper claims that this is easily feasible through generating synthetic data, but there is no evidence to back this up.
- Please clarify terminology (“functional”), fix notation clashes (d for distance vs. sample count), and address the noted language/editing issues.

---

> ### Author Response · Authors · 2025-12-03
>
> We are very grateful for your time and effort in providing your insightful comments. Your feedback helps us to substantially improve the paper.
>
> Weakness: The theoretical component is thin and rests on strong assumptions that limit real-world applicability. Methodologically, the approach largely assembles known components (SAM2/YOLO for masks, DINOv3 features, transformer on features, LLM-generated priors), and novelty resides mainly in the particular Bayesian factorization. Ablations quantifying the contribution of each choice are limited. The likelihood is trained from only seven safe-only videos, and the large count of “examples” stems from frame and pairing combinatorics rather than independent, diverse contexts. It is unclear how well the method would generalize to other scenes.
>
> The assembly of known components is typical in risk estimation in robotics [3, 1, 5], especially well-known
> foundational models such as SAM/YOLO/DINO/LLMs/VLMs. We do not perform ablations on the
> choice of models because (1) these models are standard and popular foundational models and (2) these
> models are necessary for the functionality of our risk framework. In addition, we seek to propose a risk
> framework, rather than any single model; thus, users can use our framework but swap out our choice of
> models in the future when stronger models are created. Succinctly, we simply show that the aggregation
> of these models can reasonably reason about risk and return risk images from input RGB-D images that
> are human aligned, while LLMs/VLMs cannot (as seen in our new experiment in Section 6.2).
>
> The fact that the likelihood is trained from 7 short-form, safe-only videos is a boon, as the likelihood serves
> to finetune the risk values for only a small subset of objects that a robot might see in a single deployment.
> Thus, quick and simple human demonstrations are preferable. The risk model is not intended to be a
> foundational model, though we believe our proposed risk framework is amenable and scalable for a risk
> foundation model. This fact is emphasized in Remark 3.
>
> In terms of scene generalization, the risk model is expressly zero-shot due to the fact that the model
> reasons over objects (encoded by DINOV3 semantic embeddings) rather than entire images. The images
> in our qualitative figure (Figure 4) are all in environments that were never seen in the training data used
> to regress the likelihood. In addition, many of these objects in these scenes were never in the training
> data. We include a snippet of the main text here for your convenience (Section 6.1):
>
> Quote: Many of the objects present were not in the training datasets, which speaks to the generalization
> of the risk framework. For example, the box cutter, keyboard, umbrella were not in the human
> demonstration
>
> and at the end of Trajectory Optimization Section (Section 6.3):
>
> Quote: Certain objects like the drone were never seen in either the likelihood training dataset or the
> LLM output.
>
> DINOv3 allows the model to recognize similar objects to the training data, and is also generalizable to
> related objects (e.g. a shear and a pair of scissors).
>
> [1] Lukas Brunke, Yanni Zhang, Ralf R ¨omer, Jack Naimer, Nikola Staykov, Siqi Zhou, and Angela P Schoellig.
> Semantically safe robot manipulation: From semantic scene understanding to motion safeguards. IEEE
> Robotics and Automation Letters, 2025.
>
> [3] Kensuke Nakamura, Lasse Peters, and Andrea Bajcsy. Generalizing safety beyond collision-avoidance via
> latent-space reachability analysis. 2025.
>
> [5] Leonardo Santos, Zirui Li, Lasse Peters, Somil Bansal, and Andrea Bajcsy. Updating robot safety represen-
> tations online from natural language feedback. In Proceedings of the IEEE International Conference on Robotics
> and Automation (ICRA), 2025.

---

> ### Author Response · Authors · 2025-12-03
>
> Weakness: The key assumption that the distribution of distances does not depend on the context and the normalization factor can be left out, justified by “arbitrary” distance distributions or by experimental design, is not credible for most scenarios, where distances and motion are heavily context-, affordance-, and task-dependent. Researchers designing the distribution of distances for experiments will always choose distributions based on the task.
>
> Response: We justify the independence of the distances based on two related points. First, in manipulation, the manipulated object can be put anywhere in the workspace, so there is no preference toward certain distances Risk Fields 10 for certain objects. This holds for commonly manipulated objects. Second, as mentioned in our response to another reviewer, we argue that any distance bias for a particular task is primarily due to the implicit need to be safe. Then, this distribution, conditioned on being safe and the semantics, is the likelihood and not the normalization factor. We have emphasized this point in the main text (between Remark 1 and 2),
> but displayed here for your convenience:
>
> Quote: Why should the distances be independent of the semantics? In manipulation, a robot can move a manipulable object anywhere in the workspace. Thus, without conditioning on the policy or
> the task, we should not assume that the distribution of distances should be biased for one set of
> semantics or another. It is true that objects in reality do occur with other objects, or away from
> other objects, thereby inducing a distance distribution. However, we argue that this observation
> does not preclude the marginal distance distribution from being independent of semantics. In
> fact, the real life distance distributions are conditional, induced by a task or the implicit need to
> be safe. For example, plates are typically placed near to knives and stoves because they are used
> in kitchen-related tasks. In the same vein, books are typically not stored in the kitchen due to
> water damage. Note that the distance distribution conditioned on safety is the likelihood. Since
> the marginal distribution is not conditioned on task or safety, we find that the independence
> assumption is palatable for tractability reasons.

---

> ### Author Response · Authors · 2025-12-03
>
> Weakness: The central claim that risk is non-decreasing with distance is often violated in practice. Safety semantics go well beyond distances as orientation, velocities, occlusions, containment, support relations etc may all play a role. Theorem 1 amounts to a monotonicity/ranking statement under monotone mappings and adds little substantive guarantee. The introduction’s suggestion that the method “leverages guarantees from rigorous control-theoretic approaches” overstates what is proven. There is no guarantee for anything in the pipeline, especially since the inputs for calculating the semantic-metric Bayesian risk fields come from black box neural networks.
>
> Response: We agree that in practice, safety should take into account various state information beyond distances.
> However, this does not mean the monotonicity of risk with distance is violated. A violation of monotonicity would mean that increasing the distance of two objects could increase the risk. This is atypical
> in common scenarios where manipulation is concerned. For example, moving a knife away from people
> will always be safer, and likewise with a glass of water and a laptop.
>
> We disagree with the statement that the ranking statement has little guarantee. The ranking statement
> says that, at the same distance, two different object interactions can be directly compared using the posterior risk value without having to know the evidence term. This adds substantial interpretability to the generated risk images.

---

> ### Author Response · Authors · 2025-12-03
>
> Weakness: The treatment of interchangeability is also questionable. Risk is framed as permutation-invariant in parts of the model, yet many pairwise risks are asymmetric with respect to which object is manipulated (e.g., moving water above a laptop versus moving a laptop near a stationary glass). This asymmetry should be reflected in how likelihoods and priors are conditioned.
>
> Response: While we agree that the asymmetry is a more natural way to capture risk, we believe that the interchange-
> ability assumption can still go far. First, the primary reason for interchangeability is to significantly cut
> down on the number of data points needed to learn risk. Second, if we only consider how two particular
> objects interact with each other, then moving the first object closer to the second is identical to moving
> the second towards the first, just in different reference frames. In the given example, moving the water into the laptop can cause the water to spill on the laptop due to a cup-initiated collision. Similarly, if the laptop is moved into the glass of water, it can also spill onto the laptop due to a laptop-initiated collision.

---

> ### Author Response · Authors · 2025-12-03
>
> Weakness: The prior relies on LLM ratings that are themselves subjective and prompt-sensitive. The paper notes some manual rules (e.g., forcing tabletops to zero risk), suggesting non-trivial tuning for each scenario. Important implementation details are underspecified: how 3D distances are computed robustly from RGB-D streams (calibration, missing depth, occlusions), how the “straight-line” path is justified in clutter with obstacles unrelated to the focal pair’s semantics, runtime to generate risk fields, and how thresholds for viability-to-radius conversion are selected.
>
> Response: The prior indeed relies directly on LLM ratings, which may not be human-aligned and are indeed prompt-
> sensitive. The risk framework uses the likelihood to finetune the prior in order to retrieve risk values that
> are more human-aligned. These messages are emphasized now in Remark 3 and 4. As for prompt-
> sensitivity, we do not establish any other rules beyond forcing the tabletop to be zero risk and risk be-
> tween two of the same object to be zero. These rules were established before any tests were run. All our
> experiments use the same prior, likelihood, and thus the same posterior model. In addition, these models
> are trained before test time and were asked to zero-shot estimate risk in novel scenarios.
>
> The 3D distances are simply retrieved from the Realsense/ZED depth channel from the input RGB-D im-
> age. This depth is directly used to query the prior, likelihood, and posterior. No calibration is performed,
> and the camera intrinsics are not used. Missing depth values appear in the risk image as black pixels, in-
> dicating no information. Since our models operate per frame, all RGB-D pixels in that frame are observed
> (i.e. there are no occlusions).
>
> The purposes of the likelihood data collection process is to have a safe (no dangerous interactions), simple
> data collection process that can be completed quickly on the required set of objects. Hence, the two object
> scenario is the simplest strategy to collect the required data, so the straight-line path is justified. In this
> work, the claim is not that we can learn preferences for risk from real-world human demonstrations (e.g.
> with clutter), but simply that humans can spend very little time to collect the necessary data to train the
> likelihood model. Scaling the data collection process in order to train a model more akin to a foundation
> model is planned for future work.
>
> The total time to compute a posterior risk image from an input RGB-D image is about 2.75 seconds,
> depending on the resolution of the input image. The prior and likelihood independently compute their
> risk images, and their pixel-wise product is the posterior risk image.
>
> For our hardware experiments, a threshold of α = 0.1 is chosen. 0.1 is a low value such that objects with
> low viability/high risk would achieve the maximum radii of 0.25 meters. Meanwhile, moderate objects
> could have a sizable radii, and safe objects would have very small radii.

---

> ### Author Response · Authors · 2025-12-03
>
> Weakness: On evaluation, the baselines underperform strikingly (0/33 top-1 preferences for both VLA and diffusion policy), which raises concerns about setup, training parity, and metric sensitivity. Human ratings of “riskiness” are inherently subjective. The paper would benefit from objective proxies (e.g., collision/near-miss rates under perturbations, task success under constraints, or compliance with specified semantic rules). Claims of generalization to unseen objects/contexts are not rigorously substantiated beyond qualitative figures. Distribution shift tests, cross-scene generalization, and robustness to prior misspecification are missing.
>
> Response: In fact, the baselines are given more human data for training than to our method. About an hour of human
> tele-op were used to train the VLA and diffusion policy, compared to just a few minutes for a human to collect data for the likelihood. The videos of all methods in operation can be seen in our website (https://riskbayesian.github.io/bayesian_risk/). As can
> be seen from the videos, the scenarios for all the methods are the same. The results definitively answer the
> question of whether the state-of-the-art robot policies can implicitly encode risk from risk-free training
> data, and the answer is no.
>
> The human ratings for riskiness are subjective, but this is precisely the objective of the work: to create
> a model that closely aligns with human notions of risk. To that end, we have significantly increased the
> number of participants (N = 14) for rating the trajectories to boost statistical significance. While the
> trends are still the same, with our risk model producing trajectories that are well-aligned with human
> preferences. In the same vein, the SOTA robot policies are poorly aligned, but are not 0 (meaning in a few
> scenarios, human raters did prefer these trajectories over others).
>
> Most objective proxies do not capture human alignment the same way human ratings do. However, we
> do have an objective metric (i.e. the Dynamic Time Warping metric to the human tele-op trajectory),
> which measures how far trajectories produced by other methods are to the human tele-op. In fact, the
> rating results show that our method produces trajectories that are as good as the human tele-op, which
> shows good human alignment.
>
> As mentioned in the response to another reviewer, our risk model is not the same as collision avoidance.
> For semantic safety, collision avoidance is insufficient. For example, manipulating a knife very close
> (but not touching) a human is collision avoidant, but not at all safe. In our hardware experiments, all
> trajectories achieve the simple task of manipulating the object from start to goal locations.
>
> We do not believe compliance with specific rules is a good metric because (1) the definition of these rules
> are again subjective and (2) no methods explicitly encode semantic rules (beyond the two stated earlier),
> unlike [1]. Gauging performance based on these rules would suggest that semantic risk could be easily
> decomposed into interpretable, easy-to-formulate constraints, which is simply not the case.
>
> In fact, all environments shown were not the one in which the likelihood training data was collected;
> hence a vast majority of objects have never been seen before. Less than 10 objects were used in the like-
> lihood training data. Therefore, all experiments are essentially zero-shot queries of the posterior model.
> Specific instances of out-of-distribution objects are highlighted in an earlier response.
>
> [1] Lukas Brunke, Yanni Zhang, Ralf R ¨omer, Jack Naimer, Nikola Staykov, Siqi Zhou, and Angela P Schoellig.
> Semantically safe robot manipulation: From semantic scene understanding to motion safeguards. IEEE
> Robotics and Automation Letters, 2025

---

> ### Author Response · Authors · 2025-12-03
>
> Weakness: Overall the paper is decently organized and the language is fair, but there are a several places where it requires polishing.
>
> • On line 236 it’s hard to understand what is meant by safe only human demonstrations.
>
> • Other sections are missing words, contain overly long sentences, or are grammatically incorrect (e.g. line 238, 263, 384, 421).
>
> • Active citations are being used as passive ones at line 431, 451, 456.
>
> Response: Safe-only human demonstrations is defined as demonstrations where all objects within the demonstration
> are safe and no damage or undesired behavior occurs. Additional detail on the data collection process
> can be found in Appendix A, but is included here for your convenience:
>
> Quote: The human demonstrator is only instructed to grab an object and move it around another stationary object in a natural, non-risky way from a starting point to a goal point. For the likelihood data, the start and goal points are implicitly defined by where the person picks up and
> puts down the grasped object. For the tele-operated baseline data, the tele-operator similarly moves a pre-grasped object in a natural, non-risky way from a pre-defined start pose to a pre-
> defined goal location.
>
> Requiring only safe demonstrations is advantageous to keep robots and humans healthy and functioning.
> It is common to demonstrate unsafe behavior [3] in order to avoid these behaviors. However, we show
> that our risk framework can quantify risk even when given only safe demonstrations while exhibiting
> strong alignment with human preferences.
>
> Thank you for your comments. We have revised the paper to be more succinct and readable. We have
> also changed the active citations to proper passive ones.

---

> ### Author Response · Authors · 2025-12-03
>
> Weakness: There are terminology and notation issues:
>
> • “Functional” is misused. A functional by definition refers to a mapping from functions as input to some scalar value. In the paper, the “likelihood functional” maps vectors to function parameters.
>
> • The symbol d is overloaded for both distance and the number of frames/samples.
>
> Response: We have revised the main text, referring to the likelihood only as a “function”. We now use L to represent
> the number of samples, reserving d for distance.

---

> ### Author Response · Authors · 2025-12-03
>
> Question: What concrete safety guarantees does the framework provide? Beyond the monotonic ranking lemma, can any bound or certification (e.g., chance constraints under model uncertainty) be stated for the induced trajectories?
>
> Response: Thank you for the great question. The ranking theorem, which allows different interactions to be properly compared if both interactions occur at the same distance, instills interpretability into the risk images,
> which is crucial for image-based planners to yield reasonable paths. Besides that, trajectories generated
> by classical trajectory optimizers (like the one used in our hardware experiments) contain addition safety
> guarantees. For example, given the geometry of the scene (represented as a point cloud) and the asso-
> ciated posterior for each point, we set an acceptable risk threshold α. Subsequently, the unsafe set of
> positions becomes a union of balls of varying radii around each point in the point cloud. Collision avoid-
> ance against a union of balls is a well-studied problem. Thus, all points on the trajectories produced are
> guaranteed to below the set risk threshold. Broader guarantees about the model uncertainty (e.g. about
> objects not seen in the training data) are harder to formulate, but planned for future work in the form of
> statistical guarantees.

---

> ### Author Response · Authors · 2025-12-03
>
> Question: How sensitive are results to the distance-independence assumption? If $P(\hat{d} < d | \phi)$
> depends on $\phi$ , what changes in the estimator or the training objective, and does Theorem 1 still hold?
>
> Response: The authors believe the reviewer is mistaken. The normalization factor cannot be calculated without
> having access to unsafe data. The distance-independence assumption is necessary to obtain a risk model
> that is computable.
>
> If the distance is dependent on the semantics, then Theorem 1 does not hold. The rankings can flip. Fur-
> thermore, if the normalization was comptuable, then Theorem 1 and Corollary 1 are moot, as we would
> simply use the absolute probability of being safe as the risk metric, rather than the relative probability.
> Nothing in the framework changes, as the prior and likelihood remain the same. The only difference
> would be that the risk metric would be the product of these two models, divided by the normalization
> (i.e. actual Bayes’ Rule), rather than just the product with no division.

---

> ### Author Response · Authors · 2025-12-03
>
> Question: How do you incorporate non-distance state variables (velocity, orientation, containment/support, spillage direction) that can dominate semantic risk? Can the likelihood be extended to a richer metric space while preserving smooth CDF parameterization?
>
> Response: In this current iteration of the work, we do not consider other state variables beyond distance for simplicity and tractability. While these other variables are important for semantic risk, we believe distance is the largest factor when considering risk at macroscopic distances.
>
> While it may be possible to extend the risk framework to handle additional variables (and possibly even latent variables), we believe this will require substantial data and data reformatting. First, the input space will be much larger. Second, if non-distance variables are recorded, it is not as clear how this impacts the way humans manipulate objects around other objects. For distances, it is natural: low risk interactions exhibit low distances, while high risk interactions exhibit large distances.

---

> ### Author Response · Authors · 2025-12-03
>
> Question: How is the LLM prior calibrated against human preferences beyond prompt engineering? Can you quantify inter-rater reliability, prior uncertainty, and posterior sensitivity to prior misspecification?
>
> Response: The prompt specified by humans is a tractable and efficient mechanism to enforce broad rules to estimate risk. Simultaneously, these rules can be applied to large number of objects. In order to "calibrate" the model to align with human preferences, the likelihood trained on human data modulates the risk estimate.

---

> ### Author Response · Authors · 2025-12-03
>
> Question: Have you evaluated on unseen scenes, object categories, and camera viewpoints with held-out environments? The paper claims that this is easily feasible through generating synthetic data, but there is no evidence to back this up.
>
> Response: All experiments require the model to estimate risk in a zero-shot manner. In our paper, we now note all the objects in the object LUT (Appendix B), and note in the text when objects in the image are not in the LUT (described in an earlier response). Note that many different objects from the training data (both the likelihood and prior data) can be seen in the experiments. Even for objects in the same category, objects seen in the experiments are different than those seen in training (e.g. different shape, color). Note that since the risk estimator operates on DINO features, the risk model is potentially as generalizable as DINO is. Since DINO is trained on a large corpus of images, it is very generalizable.

---

> ### Author Response · Authors · 2025-12-03
>
> Question:Please clarify terminology (“functional”), fix notation clashes (d for distance vs. sample count), and address the noted language/editing issues.
>
> Response: We appreciate the reviewer's detailed reading of the paper. We have made the appropriate changes to rectify these concerns.

---

### Official Review · Reviewer_P1Bd · 2025-10-31

**Soundness:** 2
**Presentation:** 3
**Contribution:** 2
**Rating:** 2
**Confidence:** 3

**Summary:**

This paper proposes a novel framework to learn a model of human-like, contextual safety for robot manipulation. The key idea is to formalize risk using a Bayesian formulation, where the posterior risk is proportional to a likelihood and a prior. The framework combines these two components to produce pixel-dense risk maps for a given scene and manipulated object. The authors conduct experiments qualitatively and quantitatively.

**Strengths:**

1. The core Bayesian decomposition is an elegant way to formalize intangible risk. It provides a clear and interpretable separation between behavior learned from observation and common sense knowledge.
2. A major strength is the data-collection strategy. The framework learns without requiring any unsafe demonstrations. The Likelihood is learned from safe-only human videos and the Prior is generated by a VLM.
3. This paper leverages a suite of modern foundation models to build its system. The fact that this framework's risk field, when paired with a classical trajectory optimizer, produces trajectories rated as significantly safer by humans than DP and VLA policies is very promising.
4. The output is not a monolithic policy but a risk field. This field is flexible and can be used as a value function for learned policies or as a costmap for classical planners, bridging the gap between learning and classical robotics.

**Weaknesses:**

1. The entire framework relies on the critical assumption that the evidence term is independent of the semantic context. This assumption is necessary to avoid computing intractable term. However, this seems to contradict the paper's own premise. Is it really true that the general distribution of distances between objects is independent of their semantics? Humans are likely to behave differently (and thus create different distance distributions) around a `knife` vs. a `teddy bear`, even in safe scenarios. This core assumption needs more justifications.
2. The Likelihood model, which is meant to capture all of human demonstrated behavior, is trained on only 7 videos. While this is processed into 648000 examples, it's still a tiny and low-diversity sample of human behavior. This makes the claims of generalization to novel contexts less credible, the strong performance is likely heavily reliant on the DINOv3 features rather than a richly learned likelihood model.
3. The Prior model is not a learned generative model of risk but a massive, pre-computed lookup table. A VLM is prompted to generate ~60k pairwise object ratings, and the model uses DINOv3 features to find the closest object in this table to get its risk score. This approach has two key limitations:
   - It doesn't generalize to objects truly unseen in its 300-object list, it just finds the nearest neighbor.
   - It cannot scale to combinatorial, N-way interactions (e.g., risk of `water` + `laptop` + `person`), as the LUT size would explode.
4. Results in Table 1 relies on trajectory ratings from a human rater. For a subjective, preference-based metric like `riskiness`, a single rater is a very small sample size and insufficient to draw general conclusions.
5. The paper would be significantly stronger if it included experiments in a standardized, reproducible simulation environment. This would allow for a much larger scale of quantitative testing and a fairer comparison against other methods.

**Questions:**

1. Could the authors provide a stronger justification for the key assumption that the evidence term is independent of the context? This seems to be the weakest link in the paper's theoretical foundation.
2. In Figure 4, the Likelihood map for the Cup (row 1) already seems to assign a high risk to the Laptop. The Likelihood is trained on 7 safe human demos. Did this small dataset just happen to include a `cup near laptop` demonstration? If so, doesn't this blur the clean separation of the metric Likelihood (learning from demos) and the semantic Prior (common sense)? If not, why is the Likelihood high for the laptop?
3. How does the Prior model (the LUT) handle a truly novel object that is not one of the ~300 objects queried from the VLM? Does it just default to the nearest DINOv3 neighbor, and how does this affect the risk estimate? Furthermore, how could this LUT-based approach ever scale to 3-way or N-way interactions (e.g., the risk of a `knife` near a `person` and a `fragile object`)?
4. Can the authors confirm that only one human rater was used for the trajectory preferences in Table 1? If so, this is a significant limitation. Are there plans to expand this to a more robust user study with multiple raters to validate these subjective results?
5. To improve the reproducibility and scalability of the evaluation, have the authors considered validating their framework in a standardized simulation benchmark? This would allow for more extensive quantitative comparisons and a clearer analysis of how the risk field performs against other safety-oriented planners.

---

> ### Author Response · Authors · 2025-12-03
>
> We are very grateful for your time and effort in providing your insightful comments. Your feedback helps us to substantially improve the paper.
>
> Weakness: The entire framework relies on the critical assumption that the evidence term is independent of the semantic context. This assumption is necessary to avoid computing intractable term. However, this seems to contradict the paper's own premise. Is it really true that the general distribution of distances between objects is independent of their semantics? Humans are likely to behave differently (and thus create different distance distributions) around a knife vs. a teddy bear, even in safe scenarios. This core assumption needs more justifications.
>
> Response: We thank the reviewer for the in-depth question. Our reason for the semantic independence of distances (regardless of whether the distances are safe or unsafe) is that, for a  manipulation task, a robot can manipulate an object around another object at any distance within workspace limits. For example, without any safe/unsafe conditioning, a manipulator can plausibly bring a knife as close and as far away to a human as it can a teddy bear. We would also like to clarify the example the reviewer provided (humans behavior differently around a knife vs. a teddy bear). Indeed, this is a valid phenomenon, but does not explicitly have a bearing on the semantic independence of the distance distribution. What the example captures is exactly the likelihood, the distance distribution conditioned on the semantics and the fact that the distance is safe. It is precisely the need to be safe that humans behave differently around the knife vs. a teddy bear. We have added a similar explainer in the main text between Remark 1 and 2. We include it here for the reviewer's convenience:
>
> Quote:
> Why should the distances be independent of the semantics? In manipulation, a robot can move a
> manipulable object anywhere in the workspace. Thus, without conditioning on the policy or the task,
> we should not assume that the distribution of distances should be biased for one set of semantics or
> another.
> It is true that objects in reality do occur with other objects, or away from other objects, thereby
> inducing a distance distribution. However, we argue that this observation does not preclude the
> marginal distance distribution from being independent of semantics. In fact, the real life distance
> distributions are conditional, induced by a task or the implicit need to be safe. For example, plates
> are typically placed near to knives and stoves because they are used in kitchen-related tasks. In the
> same vein, books are typically not stored in the kitchen due to water damage. Note that the distance
> distribution conditioned on safety is the likelihood. Since the marginal distribution is not conditioned
> on task or safety, we find that the independence assumption is palatable for tractability reasons.

---

> ### Author Response · Authors · 2025-12-03
>
> Weakness: The Likelihood model, which is meant to capture all of human demonstrated behavior, is trained on only 7 videos. While this is processed into 648000 examples, it's still a tiny and low-diversity sample of human behavior. This makes the claims of generalization to novel contexts less credible, the strong performance is likely heavily reliant on the DINOv3 features rather than a richly learned likelihood model.
>
> Response: We would like to clarify that the likelihood is not intended to capture all human behavior. The reviewer is correct in identifying the low-data regime that our method resides in, and the generalization power of DINO. We believe these are strengths of our method, rather than weaknesses. We have revised the text to be more specific (Remark 3 and 4), stating the following: (1) the LLM-derived prior exhibits broad generalization due to the scalability of the number of objects it can label, (2) the likelihood finetunes the prior solely on objects the model may see at deployment, and (3) DINO allows generalization beyond the objects in the likelihood data (e.g. a "cup" and a "mug" have similar DINO features). This last point can be seen in the updated Figure 4, as the box cutter was never in the human demonstrations. In addition, we have highlighted that the ability to train a small model quickly on few natural human demonstrations is a boon when compared to the data collection process required to train sub-optimal policies like diffusion and VLAs (at least one hour). However, our framework still allows for future scalability to wider datasets that aligns closer to the capabilities of a foundation model.

---

> ### Author Response · Authors · 2025-12-03
>
> Weakness: The Prior model is not a learned generative model of risk but a massive, pre-computed lookup table. A VLM is prompted to generate ~60k pairwise object ratings, and the model uses DINOv3 features to find the closest object in this table to get its risk score. This approach has two key limitations:
>
> $\cdot$ It doesn't generalize to objects truly unseen in its 300-object list, it just finds the nearest neighbor.
>
> $\cdot$ It cannot scale to combinatorial, N-way interactions (e.g., risk of water + laptop + person), as the LUT size would explode.
>
>
> Response: We believe the proposed framework is not as un-scalable as it may seem. The list of object pairs and their pairwise risk ratings are produced by the generative model in minutes, and only requires a human prompter. If more pairings are required, these can be generated much quicker than a human labeler (albeit less human-aligned, which is reason for the likelihood). The "training" process for a LUT is instantaneous, compared to minutes or hours for a deep network. Even if there were 1000 objects in the list, there would be less than 1 million keys in the LUT, which is still very efficient to query on GPU. In practice, modern image datasets with object categories are approximately around 1000 categories [1]. The use of DINO features for the lookup enables us to virtually expand the number of identifiable objects (e.g. the DINO feature for "cup" is very similar to "mug"), generalizing past the list of 300 objects. This can be seen in the robot hardware demonstrations, which can be visualized on our website (https://riskbayesian.github.io/bayesian_risk/#risk-aware-robot-navigation). Specifically, the drone (Trial 19) is not in the human demonstrations or the LLM output.
>
> To address the second point, if the semantic risk is the cause of a 3-way interaction, then indeed a naive LUT would explode. With 1000 objects, the naive LUT would have 1 billion entries. However, this would also introduce a lot of sparsity, as many combinations would pose no risk to each other. While this was not done in our implementation for simplicity, the risk LUT could return no risk if the N-way object pair did not exist in the LUT. Note that this only applies to the risk LUT, which is a dictionary with keys being text. The object LUT is a nearest-neighbor LUT, but scales linearly with the number of distinct object classes. To give some perspective, the object LUT which converts a DINO feature to a string (e.g. "cup") contains approximately 300 object classes. The risk LUT which converts a key containing two strings (e.g. "cup" and "laptop") to a quantized risk value contains approximately 60K entries, with 47K of them being no risk. Effectively, we could have stored only 13K entries in the risk LUT and recovered identical behavior.
>
> [1] Olga Russakovsky, Jia Deng, Hao Su, Jonathan Krause, Sanjeev Satheesh, Sean Ma, Zhiheng Huang, Andrej
> Karpathy, Aditya Khosla, Michael Bernstein, Alexander C. Berg, and Li Fei-Fei. ImageNet Large Scale
> Visual Recognition Challenge. International Journal of Computer Vision (IJCV), 115(3):211–252, 2015. doi:
> 10.1007/s11263-015-0816-y.

---

> ### Author Response · Authors · 2025-12-03
>
> Weakness: Results in Table 1 relies on trajectory ratings from a human rater. For a subjective, preference-based metric like riskiness, a single rater is a very small sample size and insufficient to draw general conclusions.
>
> Response: We have significantly expanded the number of participants to $N = 14$  different human participants. In the expanded experiment, we still see that our risk model is human-aligned compared to the SOTA robot policy baselines (now Table 2). Note that in our original Table 1, we also include a non-subjective metric, which is the Dynamic Time Warping (DTW) between the different methods to the human tele-op. We show that our trajectories are quite aligned with those produced by the human.

---

> ### Author Response · Authors · 2025-12-03
>
> Weakness: The paper would be significantly stronger if it included experiments in a standardized, reproducible simulation environment. This would allow for a much larger scale of quantitative testing and a fairer comparison against other methods.
>
> Response: We have significantly strengthened the paper by including an experiment benchmarking our method as a risk value learner compared to VLMs (e.g. ChatGPT5 and Gemini3). The three methods rated the riskiness of several trajectories represented as a sequence of image frames. The quality of each method's choice of trajectory was based on human participants' agreement with these trajectories. The videos of the trajectories are included on our website (https://riskbayesian.github.io/bayesian_risk/). We find that human participants agreed with our method's determination of most and least risky trajectories, while this is not the case with the VLMs (Table 1).
>
> The choice of such an experimental design was primarily motivated by realism and lack of suitable baselines. We are not aware of any architectures that can quantify risk zero-shot from RGB images, besides VLMs. Moreover, rather than a simulation environment, which introduces a sim-to-real gap for robot tasks, we opted to use videos from real environments.

---

> ### Author Response · Authors · 2025-12-03
>
> Question: Could the authors provide a stronger justification for the key assumption that the evidence term is independent of the context? This seems to be the weakest link in the paper's theoretical foundation.
>
> Response: Thank you for your question. We have answered the question in a previous comment.

---

> ### Author Response · Authors · 2025-12-03
>
> Question: In Figure 4, the Likelihood map for the Cup (row 1) already seems to assign a high risk to the Laptop. The Likelihood is trained on 7 safe human demos. Did this small dataset just happen to include a cup near laptop demonstration? If so, doesn't this blur the clean separation of the metric Likelihood (learning from demos) and the semantic Prior (common sense)? If not, why is the Likelihood high for the laptop?
>
> Response: Thank you for your insightful question. Indeed, in our computer lab setting, we chose to include the laptop-cup object pair into the human demonstrations as we found it to be relevant for the setting. Since the prior also believes this pairing to be risky, the posterior will only see this pairing's risk magnified. Moreover, it is inevitable that the likelihood training set will share pairs with the LLM output, due to the vast number of items produced by the LLM  (Appendix B). There are also instances in which the likelihood and prior disagree, like in Row 3 (Figure 4) for the box cutter. Overall, the overall purpose of the likelihood is to just modulate the prior's risk to be more human aligned. This is now emphasized in Remark 3.

---

> ### Author Response · Authors · 2025-12-03
>
> Question: How does the Prior model (the LUT) handle a truly novel object that is not one of the ~300 objects queried from the VLM? Does it just default to the nearest DINOv3 neighbor, and how does this affect the risk estimate? Furthermore, how could this LUT-based approach ever scale to 3-way or N-way interactions (e.g., the risk of a knife near a person and a fragile object)?
>
> Response: Thank you for your question. We have answered the question in a previous comment.

---

> ### Author Response · Authors · 2025-12-03
>
> Question: Can the authors confirm that only one human rater was used for the trajectory preferences in Table 1? If so, this is a significant limitation. Are there plans to expand this to a more robust user study with multiple raters to validate these subjective results?
>
> Response: We have significantly expanded the Results section and increased the number of human participants. The precise details can be found in an earlier comment.

---

> ### Author Response · Authors · 2025-12-03
>
> Question: To improve the reproducibility and scalability of the evaluation, have the authors considered validating their framework in a standardized simulation benchmark? This would allow for more extensive quantitative comparisons and a clearer analysis of how the risk field performs against other safety-oriented planners.
>
> Response: The Results section have been greatly expanded to include benchmarks against other baselines in real scenarios, rather than simulated ones. Please see our earlier response for more specific details.

---

### Official Review · Reviewer_NSa1 · 2025-11-01

**Soundness:** 2
**Presentation:** 2
**Contribution:** 3
**Rating:** 2
**Confidence:** 4

**Summary:**

The paper proposes a Bayesian framework to quantify risk/viability as a function of context and pairwise object distance. By assuming the distribution of distances is context-independent, viability $\propto$ semantic prior of safety, $P(\text{safe}|\phi)\times$ the CDF of $p(d|\text{safe}, \phi)$. The semantic prior of safety is derived from common sense knowledge of ChatGPT-5. The conditional distance distribution $p(d|\text{safe}, \phi)$ is learned from safe-only human demonstrations.

**Strengths:**

- The paper addresses the important question of inferring semantic safety that is aligned with human preferences.
- The proposed method does that by learning from safe-only human demonstrations.

**Weaknesses:**

- While the work is motivated by semantic safety, it relies heavily on distance between objects snd collision avoidance.
- The absolute value of the learned viability in this work has no meaning, only the relative value does, which introduces difficulty in interpreting the value. This is a result of the normalization factor in Bayes inference cannot be computed. More commonly, risk is defined as the probability of failure $\in [0. 1]$
- The experimental results are weak. There is no comparison to any method of estimating risk from demonstrations or any risk-aware control policy.
    - In the first downstream application of value predictor for visuomotor policy, only qualitative results are shown. To be honest, the qualitative results do not entirely make sense. For instance, 2nd row of Figure 4, there is only medium risk (yellow) of placing the laptop on the cup. The learned likelihood map looks spare, which may be a result of training on limited data.
    - In the second downstream application of risk-aware trajectory optimization, neither of the baselines explicitly considers risk...

**Questions:**

- How is the experiment for data collection designed? What are the human demonstrator instructed to do? How much data is sufficient?

---

> ### Author Response · Authors · 2025-12-03
>
> We are very grateful for your time and effort in providing your insightful comments. Your feedback helps us to substantially improve the paper.
>
> Weakness: While the work is motivated by semantic safety, it relies heavily on distance between objects and collision avoidance.
>
> Response: We strongly believe that quantifying safety is dependent both on the semantics of the objects involved as well as the distance between these objects. For example, knives in the kitchen pose no treat to humans in the bedroom. While distance may not be the only thing that influences risk, we believe that representing risk as a spatial field is a meaningful starting point to spawn future research.
>
> While subtle, we would also like to differentiate how the presented risk framework differs from collision avoidance (e.g. [1]), which has enjoyed many years of dedicated research. One of the biggest differences is that objects of 0 risk can be collided with (e.g. moving a glass of water through a cereal box), due to the fact that risk is interpreted through a continuous value function lens, rather than a binary yes/no classifier for the object interaction. Closely related, collision avoidance is typically behavior that involves sets. Therefore, all behavior that lives outside the unsafe set are equally valid from a collision avoidance perspective. The resultant behavior is that some subset of a trajectory will live on the boundary of the unsafe set. For example, manipulating a knife very close to a human (but not touching them) may be collision avoidant, but very risky. There can also be cases where no feasible trajectory exists that accomplishes the required task, which can severely degrade the performance and usefulness of a robot if this phenomenon happens too often.
>
> While risk and collision avoidance are different, we emphasize that our risk framework can be complementary with set-based constraints. Our risk framework produces continuous values which can be used in the cost, which is separate from any trajectory constraints.
>
> [1] Lukas Brunke, Yanni Zhang, Ralf R ¨omer, Jack Naimer, Nikola Staykov, Siqi Zhou, and Angela P Schoellig.
> Semantically safe robot manipulation: From semantic scene understanding to motion safeguards. IEEE
> Robotics and Automation Letters, 2025.

---

> ### Author Response · Authors · 2025-12-03
>
> Weakness: The experimental results are weak. There is no comparison to any method of estimating risk from demonstrations or any risk-aware control policy.
>
> $\cdot$ In the first downstream application of value predictor for visuomotor policy, only qualitative results are shown. To be honest, the qualitative results do not entirely make sense. For instance, 2nd row of Figure 4, there is only medium risk (yellow) of placing the laptop on the cup. The learned likelihood map looks spare, which may be a result of training on limited data.
>
> $\cdot$ In the second downstream application of risk-aware trajectory optimization, neither of the baselines explicitly considers risk...
>
> Response: We have significantly expanded our Results section as well as elaborated on our experimental details. As
> stated in another reviewer’s response, we find the state-of-the-art in risk estimation for robotic tasks (like
> planning) to be insufficient. First, we are not aware of any methods that estimate risk zero-shot from RGB
> images; other methods require manual input [1] or are evaluated in very similar environments to those
> they are trained on and require unsafe demonstrations [3]. Second, there exists a human tele-op baseline.
> This baseline is the defacto ”best” policy by definition. Our results (Table 2) show that the posterior is very
> close to the ”best” policy in riskiness, with almost no hand-holding and generalizes zero-shot compared
> to existing method. Finally, it is unclear in the literature whether SOTA robot policies that learn from
> safe tele-op demonstrations could implicitly encode risk, especially since diffusion policy is known to
> memorize trajectories [2] and VLAs use a VLM backbone, just as our risk framework does. Therefore, it
> could have been possible that these policies are risk-aware. However, our results in Table 2 demonstrate
> that they do not.
>
> We have also included a new experiment, with baselines that can be used as risk-aware value signals (and
> hence policies). Specifically, we compared ChatGPT5, Gemini3, and our posterior model in trajectory risk
> value estimation by rating proposed trajectories (represented as several image frames) from least risky to
> most risky. Human participants also rated these trajectories from least to most risky. Table 1 tallies the
> number of agreements between the method and the human participants. Not only do these VLMs have
> structural limitations (like a context window of 10 images per prompt) that limit input context, but they
> are overall not as human-aligned as our risk model.
>
> In response to the reviewer’s concerns about Figure 4, the likelihood can down-weight risk from the prior.
> Indeed, the likelihood image is typically sparse, reflecting the small training dataset size. The reason for
> a smaller dataset size is so that users can quickly collect data and train a small model (about 54 MB) that
> modulates the prior on only objects that are relevant at test time. We emphasize this fact in Remark 3.
>
> [1] Lukas Brunke, Yanni Zhang, Ralf R ¨omer, Jack Naimer, Nikola Staykov, Siqi Zhou, and Angela P Schoellig.
> Semantically safe robot manipulation: From semantic scene understanding to motion safeguards. IEEE
> Robotics and Automation Letters, 2025.
>
> [2] Chengyang He, Xu Liu, Gadiel Sznaier Camps, Guillaume Sartoretti, and Mac Schwager. Demystifying
> diffusion policies: Action memorization and simple lookup table alternatives, 2025. URL https://arxiv.or
> g/abs/2505.05787.
>
> [3] Kensuke Nakamura, Lasse Peters, and Andrea Bajcsy. Generalizing safety beyond collision-avoidance via
> latent-space reachability analysis. 2025.

---

> ### Author Response · Authors · 2025-12-03
>
> Question: How is the experiment for data collection designed?
>
> Response: There are two instances of data collection: for training the likelihood model and for training the baselines (i.e. diffusion policy and GR00T). Both are simple instances of tabletop manipulation of one object around another, or possibly on top if they are not risky. For the likelihood model, we only trained on videos of trajectories involving 7 distinct object interactions. Each video was typically 30 seconds and no more than two minutes. For the baselines, a human tele-operated a manipulator holding an object and navigated it around another object, for about 30 different object interactions. The baseline data collection process took about an hour. We have added these details to Appendix A.

---

> ### Author Response · Authors · 2025-12-03
>
> Question: What are the human demonstrator instructed to do?
>
> Response: The human demonstrator is only instructed to grab an object and move it around another stationary object in a natural, non-risky way from a starting point to a goal point. For the likelihood data, the start and goal points are implicitly defined by where the person picks up and puts down the grasped object. For the tele-operated baseline data, the tele-operator similarly moves a pre-grasped object in a natural, non-risky way from a pre-defined start pose to a pre-defined goal location. Again, we have added these details to Appendix A.

---

> ### Author Response · Authors · 2025-12-03
>
> Question: How much data is sufficient?
>
> Response: The data used to train the likelihood were several small-length videos from 30 seconds to 2 minutes each. The role of the likelihood in our framework is to finetune the risk model for pertinent object pairs for a task. Therefore, the data should be simple, targeted, and fast to collect. While interesting, the goal of the current risk framework is not to build a risk foundation model that is trained on a large corpus of data; we do hope that researchers extend our proposed framework to build such a model. We again emphasize this message in Remark 3.

---

> ### Author Response · Authors · 2025-12-03
>
> Weakness: The absolute value of the learned viability in this work has no meaning, only the relative value does, which introduces difficulty in interpreting the value. This is a result of the normalization factor in Bayes inference cannot be computed. More commonly, risk is defined as the probability of failure $\in [0,1]$.
>
> Response: This is an insightful observation, and is intended by design. From a human perspective, quantifying the risk between two objects is subjective; only the ordering of object pairs (e.g. a cup and a laptop is more risky than a cup and a paper plate) is pertinent, and can be gotten from the relative value (i.e. the posterior). We also note that because the risk value is defined as the product of two valid probabilities, the risk value is also bounded between 0 and 1; the only downside is that the risk is not a valid probability distribution.

---

### Official Review · Reviewer_fQCW · 2025-11-03

**Soundness:** 3
**Presentation:** 3
**Contribution:** 3
**Rating:** 6
**Confidence:** 3

**Summary:**

This paper proposes Semantic-Metric Bayesian Risk Fields, a novel framework for learning contextual, spatially-varying risk from human demonstration videos using a Bayesian formulation. The prior is derived from a vision-language model (VLM) and large language model (LLM), while the likelihood is learned from object interactions in human demonstrations. The resulting pixel-dense risk maps can be used for robot planning and trajectory optimization. The authors demonstrate the model’s generalization and utility across several downstream tasks.

**Strengths:**

* The paper is well-written and conceptually clear, with strong motivation for modeling semantic safety.
* The Bayesian formulation is elegant and aligns well with human-like reasoning about risk.
* The integration of VLM features and LLM-derived priors is creative and enables generalization to unseen contexts.
* The image processing pipeline and use of Bézier curve fitting for CDFs are technically practical.

**Weaknesses:**

* While the framework is novel in its composition, many components (e.g., risk from demonstrations, VLM features, LLM priors) are adapted from existing ideas.
* The theoretical contributions (e.g., viability consistency, risk consistency) are intuitive and not particularly deep.
* The dataset used for likelihood regression is small, and the evaluation lacks rigorous quantitative comparisons to baselines.
* The prior fitting relies heavily on LLM outputs, which may not always align with human preferences without careful calibration.

**Questions:**

* How sensitive is the model to inaccuracies in the LLM-derived prior?
* Can the authors provide more quantitative comparisons to existing risk-aware planning methods?
* How well does the model perform in cluttered or ambiguous scenes where object semantics are less clear?

---

> ### Author Response · Authors · 2025-12-03
>
> We are very grateful for your time and effort in providing your insightful comments. Your feedback helps us to substantially improve the paper.
>
> Weakness: While the framework is novel in its composition, many components (e.g., risk from demonstrations, VLM features, LLM priors) are adapted from existing ideas.
>
> Response: There is a strong precedent for using these components in risk learning for robotics tasks  [1, 2, 3]. What differentiates our work is the ability to combine common sense knowledge from LLMs, generalizable semantic feature extraction from DINO, and safe-only human demonstrations to zero-shot estimate risk from RGB images. Additional functionality includes extracting risk images, reasoning images, and value signals. We believe this constitutes as a novel contribution, despite leveraging existing models and ideas.
>
> [1] Kensuke Nakamura, Lasse Peters, and Andrea Bajcsy. Generalizing safety beyond collision-avoidance via
> latent-space reachability analysis. 2025.
>
> [2] Lukas Brunke, Yanni Zhang, Ralf R ¨omer, Jack Naimer, Nikola Staykov, Siqi Zhou, and Angela P Schoellig.
> Semantically safe robot manipulation: From semantic scene understanding to motion safeguards. IEEE
> Robotics and Automation Letters, 2025.
>
> [3] Leonardo Santos, Zirui Li, Lasse Peters, Somil Bansal, and Andrea Bajcsy. Updating robot safety represen-
> tations online from natural language feedback. In Proceedings of the IEEE International Conference on Robotics
> and Automation (ICRA), 2025.

---

> ### Author Response · Authors · 2025-12-03
>
> Weakness: The theoretical contributions (e.g., viability consistency, risk consistency) are intuitive and not particularly deep.
>
> Response: We believe the intuitiveness of the theory is actually a strength and not a weakness. The prior, derived straight from a LLM, is not necessarily human-aligned. Bayes' Rule allows us to readily finetune the existing model with human data by creating another model (the likelihood) that modulates the prior to create the posterior. The straightfowardness of the framework also allows users to be spun-up very quickly should they elect to use other priors or human data. In regards to the Theorem and Corollary, we find them to be crucial in retrieving a risk metric that is interpretable. If the sorting were not preserved between the absolute and relative probabilities, then the relative risk metric cannot be used as a proxy for the probability of safety.

---

> ### Author Response · Authors · 2025-12-03
>
> Weakness: The dataset used for likelihood regression is small, and the evaluation lacks rigorous quantitative comparisons to baselines.
>
> Response: In response to insufficient quantitative results, we have augmented the Results section with an additional experiment and expanded the existing robot hardware experiment. The new experiment is dedicated to comparing the posterior's human alignment compared to SOTA VLMs (which are supposed to mimic human intelligence) in correctly ranking trajectories based on risk. 17 human participants were asked to rate these trajectories, serving as the ``oracle" (Section 6.2). Moreover, we've expanded the robot hardware experiment to 14 participants ranking the quality of the different manipulator planning methods (Section 6.3).
>
> In regards to the size of the training set for the likelihood, the intended purpose of the likelihood was to finetune the prior, which has knowledge of many more object interactions that may not be relevant for deployment. In fact, our likelihood model is incredibly small, approximately 54 MB. In short, the likelihood model is not intended to be a foundation model, but rather a model in which users could train very quickly, trained from data that could be generated very quickly from humans and is relevant to the task (i.e. tabletop manipulation of a kitchen area). The generalization capabilities of the model primarily come from the prior (and partly due to DINO), due to the expansive knowledge of the LLM. The human/task-alignment come from the human data used to train the likelihood, which is used to modulate the prior. This fact is now emphasized in Remark 3.

---

> ### Author Response · Authors · 2025-12-03
>
> Weakness: The prior fitting relies heavily on LLM outputs, which may not always align with human preferences without careful calibration.
>
> Response: We agree that the prior is strictly derived from the LLM, which may not be aligned with human preferences. This is intended. The LLM is meant to capture very broad rules (e.g. don't mix water with electronics, keep sharp objects away from tissue), but may not be specific enough for deployment. Human alignment is handled by the likelihood, which is trained from a small number of human demonstrations to accurately reflect the set of objects and specific context during deployment. This fact is now emphasized in Remark 4.

---

> ### Author Response · Authors · 2025-12-03
>
> Question: How sensitive is the model to inaccuracies in the LLM-derived prior?
>
> Response: Great question. We again emphasize that the likelihood is designed to correct alignment problems present in the prior. Qualitatively, we see in Figure 4 (Row 3) that the prior is completely blank (i.e. it thinks everything is safe), but the likelihood picks up the box cutter as risky. Consequently, the posterior also believes the box cutter is risky w.r.t the penguin. Similarly, we see that the stuffed penguin is safe with respect to the cup of water in the prior (Row 1), but is picked up in the likelihood due to its presence in the human demonstrations. Subsequently, the penguin also lights up in the posterior.

---

> ### Author Response · Authors · 2025-12-03
>
> Question: Can the authors provide more quantitative comparisons to existing risk-aware planning methods?
>
> Response: Certainly. As stated previously, we have added an additional experiment (Table 1), comparing the risk model to VLMs as a value signal. The signal can then be used to choose desirable (i.e. low risk) trajectories. We have also added more description (and expanded the pool of human participants) to the existing hardware experiment to showcase that the SOTA robot policies are trained on safe demonstrations, but show that they do not generalize to accurately quantify risk. Other methods that are risk-aware either must be trained on unsafe demonstrations [1], or require manual labor to specify safe behavior and do not work directly on images [2]. This is why, instead, we propose an experiment that uses two state-of-the-art VLMs (i.e. ChatGPT and Gemini) as value learners and show that our framework chooses less risky trajectories.
>
> [1] Kensuke Nakamura, Lasse Peters, and Andrea Bajcsy. Generalizing safety beyond collision-avoidance via latent-space reachability analysis. 2025.
>
> [2] Lukas Brunke, Yanni Zhang, Ralf R ¨omer, Jack Naimer, Nikola Staykov, Siqi Zhou, and Angela P Schoellig. Semantically safe robot manipulation: From semantic scene understanding to motion safeguards. IEEE Robotics and Automation Letters, 2025.

---

> ### Author Response · Authors · 2025-12-03
>
> Question: How well does the model perform in cluttered or ambiguous scenes where object semantics are less clear?
>
> Response: The risk model is conditioned purely on DINO features. In terms of extracting an image of DINO features from an RGB image, DINO is designed for this task, as demonstrated by the open-world visualizations on their paper [1]. However, two flaws do exist. First, our method is dependent on DINO (or another other feature extractor), so any failures with DINO will cascade into the risk model. Secondly, the risk image can be quite noisy; to tackle this, we implemented a post-processing procedure that uses SAM to propose object masks, with the risk averaged over this mask (updated Figure 4). This ensures that each object in the image is only assigned a single risk value. In practice, as seen from the success in estimate risk as a value signal (Figure 5), real scenes do not impact risk estimate quality.
>
> [1] Oriane Sim´eoni, Huy V. Vo, Maximilian Seitzer, Federico Baldassarre, Maxime Oquab, Cijo Jose, Vasil
> Khalidov, Marc Szafraniec, Seungeun Yi, Micha¨el Ramamonjisoa, Francisco Massa, Daniel Haziza, Luca
> Wehrstedt, Jianyuan Wang, Timoth´ee Darcet, Th´eo Moutakanni, Leonel Sentana, Claire Roberts, Andrea
> Vedaldi, Jamie Tolan, John Brandt, Camille Couprie, Julien Mairal, Herv´e J´egou, Patrick Labatut, and Piotr
> Bojanowski. DINOv3, 2025. URL https://arxiv.org/abs/2508.10104.

---

### Meta-Review · Area_Chair_GQ3e · 2025-12-30

**Summary:**

The paper proposes a Semantic-Metric Bayesian Risk Field framework that models robot safety as a continuous, semantics-aware spatial risk function by combining an LLM/VLM-based semantic prior with a distance-based likelihood learned from safe human demonstrations. A key strength of the work is its clear and practical formulation of semantic safety as a reusable intermediate representation. However, reviewers raised several central concerns, such as (1) the core assumption that risk can be adequately modeled as a distance-based spatial field conditioned by semantics remains not thoroughly validated; (2) the theoretical contribution is limited in depth, which provides only intuitive consistency arguments rather than control-theoretic guarantees; and (3) the likelihood model is trained from a very small number of demonstrations (seven), leaving open questions about robustness and generalization to diverse contexts despite clarifications in the rebuttal.

**Reviewer Concerns:**

The rebuttal provided thoughtful clarifications and additional context for all three major concerns, but addressed them only in a limited manner. (1) For the distance-based risk modeling assumption, the authors clarified their conceptual motivation. However, they did not provide new experiments to directly validate this core assumption, leaving the concern largely outstanding. (2) Regarding the theoretical contribution, the rebuttal tried to reframe the theory as a sanity check, but it remained concerned about the limited depth and practical implications of the theoretical results. (3) Finally, while the authors clarified how frame-level observations expand the effective data size for likelihood learning, the absence of additional experiments to demonstrate robust generalization for broader scenarios (note: seven demonstrations would be semantically related) from such limited demonstrations means that concerns about data scale and generalization remain unresolved.

**Reviewer Scores:**

The discussion may have led to slight score increases for some reviewers by clarifying the authors’ intent, but it did not address the core concerns strongly enough to change the overall evaluation. Any score changes would likely be marginal rather than a game changer.

---

### Decision · Program_Chairs · 2026-01-26

Reject